# Efficient Gradient-Based Algorithm for Training Deep Learning Models With Many Nonlinear Activations

## Abstract

This research paper presents a novel algorithm for training deep neural networks with many nonlinear layers (e.g., 30). The method is based on backpropagation of an approximated gradient, averaged over the range of a weight update. Unlike the gradient, the average gradient of a loss function is proven within this research to provide more accurate information on the change in loss caused by the associated parameter update of a model. Therefore, it may be utilized to improve learning. In our implementation, the efficiently approximated average gradient is paired with RMSProp and compared to the typical gradient-based approach. For the tested deep model with numerous stacked fully-connected layers featuring nonlinear activations on MNIST and Fashion MNIST, the presented algorithm: (a) generalizes better, at least in a reasonable epoch count, (b) in the case of optimal implementation, learning would require less computation time than the gradient-based RMSProp, with the memory requirement of the Adam optimizer, (c) performs well on a broader range of learning rates, therefore it may bring time and energy savings from reduced hyperparameter searches, (d) improves sample efficiency about three times according to median training losses. On the other hand, for a deep sequential convolutional model trained on the IMDB dataset, sample efficiency is improved by about 55%. However, in the case of the tested shallow model, the method performs approximately the same as the gradient-based RMSProp in terms of both training and test loss. The source code is provided at [...].

## 1 Introduction

### 1.1 Average Gradient

In this research, we focus on solving deep learning problems by calculating the precise influence of potential updates on the loss for each model parameter separately. Our goal is to obtain more precise information about the influence of each parameter on loss than what the gradient provides. Each potential update of a model parameter influences the locally-optimal direction of other model parameters during the same weight update, highlighting the complexity of the problem. The average gradient (defined in Appendix A), unlike the gradient, stores the accurate contribution of each model parameter to the loss delta related to a given weight update (Fig. 1; Eq. 14). Therefore, the average gradient can be utilized to efficiently minimize the loss. In this research, we propose a very fast algorithm to approximate the average gradient. We prove its approximation accuracy, validate the proof using our handcrafted metric to compare batch-loss minimization efficiency between methods, and test our method on various domains and models. Our algorithm in its current form primarily targets very deep models with many nonlinear layers.

Due to the tendency to increase model depth along with its width (Tan & Le, 2019) and the popularity of certain nonlinear activation functions, our approach may offer insights for future improvements in practical deep learning. Our primary target is to significantly improve sample efficiency, even at the cost of a moderate increase in computation time, which is essential for practical applications in fields like deep reinforcement learning or reinforcement learning from human feedback (Kirk et al., 2023). These methods have been used in popular chatbots, such as OpenAI's ChatGPT (OpenAI, 2023) and Anthropic's Claude (Kirk et al., 2023).

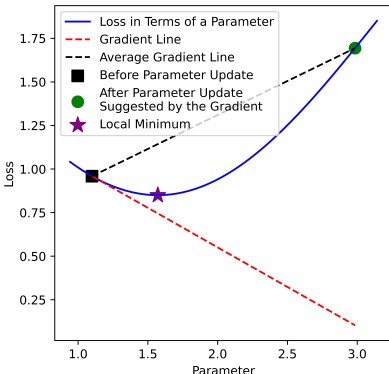 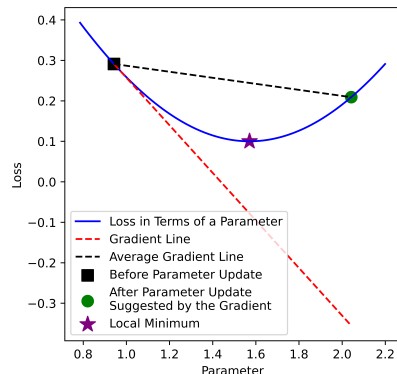

(a) *Example 1.* The average gradient suggests a different direction for updating a particular parameter.

(b) *Example 2.* If the average gradient decreases in the same direction as the gradient, it additionally provides more information about the loss landscape.

Figure 1: *Comparison of Gradient and Average Gradient.* The latter accurately reflects the influence of a parameter update on loss (as described by Equation 3, under the assumptions that $f$ represents the visualized loss function, with $x$ and $x'$ denoting the parameter values before and after an update, respectively). The plots refer to a simple case with only one parameter of the model. However, they can also be understood to present the loss contribution of a parameter during a weight update involving multiple model parameters. Appendix F presents visualizations involving two parameters.

## 1.2 GRADIENT OPTIMIZATION AND AVERAGING

Gradient optimization dominates deep learning with optimizers like Stochastic Gradient Descent (Liu et al., 2020), RMSProp (Tieleman et al., 2012), Adam (Kingma & Ba, 2014) or Nadam (Dozat, 2016). The leading algorithms for training do not change frequently over the years. However, our algorithm or its variants may be used along with first-order optimizers.

Gradient averaging is commonly used in machine learning, but in a distinct scenario than in our approach. Momentum is the running average of gradients over subsequent batches (Liu et al., 2020; Kingma & Ba, 2014; Dozat, 2016). It prevents falling into local minimums and may accelerate learning. Similarly, averaging model parameters may improve convergence and learning speed (Ruppert, 1988; Polyak & Juditsky, 1992; Merity et al., 2017; Wei et al., 2023; Sun et al., 2010), though it requires a significant amount of memory. The technique can be described as averaging a function of gradients, as the averaged model parameters over subsequent updates depend on the gradient values.

Accumulating gradients over a batch is inherent in machine learning. In practice, it is equivalent to averaging gradients computed for multiple inputs. However, this approach alone does not take into account the information about a parameter update (Fig. 1), which remains unknown during its computation. Consequently, it does not guarantee the accuracy of computing the influence of the unknown parameter update on the loss. Nevertheless, batching remains fully compatible with our method and is employed in our implementation.

Our approach is more closely related to some second-order optimization methods (Tan & Lim, 2019) rather than momentum-based or parameter-averaging techniques. This is due to the utilization of information about the curvature of a loss function during each parameter update (Fig. 1). Recently, one of the most popular algorithms for second-order optimization of neural networks is L-BFGS (Berahas et al., 2016). However, the current methods in this field are impractical for training large models due to their computational inefficiency or substantial memory requirements.

The integrated gradient, closely related to the average gradient, is used in some neural-network explainability techniques (Sundararajan et al., 2017; Khorram et al., 2021; Sattarzadeh et al., 2021). However, the approximation algorithms for the integral of the gradient used in the literature are very inefficient to compute for every parameter update of a model due to the calculation of the Riemann sum (Hughes-Hallett et al., 2021).

## 2 METHODS

### 2.1 ALGORITHM

All of the best and most popular optimizers for training large neural networks rely on the gradient. Consequently, they explicitly ignore how loss function in terms of model parameters behaves in the range between before and after a potential weight update (Fig. 1). The definition of the gradient implies, that it reflects the accurate influence on loss only for learning rates approaching to zero, which does not hold in practice. Consequently, gradient-based optimizers do not calculate the accurate influence on loss of a potential weight update, which may significantly slow down the learning of very deep models with many nonlinear operators, as our experiments show. The average gradient solves the described problem. Our algorithm efficiently approximates the average gradient, providing more reliable information on the update direction that minimizes the loss. The average gradient (contrary to the gradient) is directly proportional to the loss delta (Fig. 1; Equation 14 in Appendix B), hence it accurately describes the influence on loss of a parameter delta.

In our algorithm, given a sequential model, the average gradient is approximated and propagated according to the equation proven in Appendix B:

$$\underset{\theta_k}{\mathcal{AVG}} \nabla_{\theta_k} \ell \cong \underset{\theta_k}{\mathcal{AVG}} \frac{\partial \boldsymbol{x}_k}{\partial \theta_k} \cdot \underset{\boldsymbol{x}_k}{\mathcal{AVG}} \frac{\partial \boldsymbol{x}_{k+1}}{\partial \boldsymbol{x}_k} \cdot \ldots \cdot \underset{\boldsymbol{x}_{n-1}}{\mathcal{AVG}} \frac{\partial \boldsymbol{x}_n}{\partial \boldsymbol{x}_{n-1}} \cdot \underset{\boldsymbol{x}_n}{\mathcal{AVG}} \nabla_{\boldsymbol{x}_n} \ell \tag{1}$$

where $\ell$ is a loss function, $\theta_k$ are parameters of a layer no. $i$ and $(\boldsymbol{x}_k, \boldsymbol{x}_{k+1}, \ldots, \boldsymbol{x}_n)$ are inputs and outputs of subsequent layers of a neural network. The notation $\nabla_{\boldsymbol{x}} f$ refers to the gradient of some function $f$ for an argument $\boldsymbol{x}$, and $\frac{\partial f}{\partial \boldsymbol{x}}$ denotes the Jacobian. The average operator $\mathcal{AVG}$ of gradients or Jacobians is defined in Appendix A and aligns with intuition. The averages are aggregated with respect to the parameters of a model ($\theta_k$) or the outputs of subsequent layers ($\boldsymbol{x}_k, \boldsymbol{x}_{k+1}, \ldots, \boldsymbol{x}_n$). The average gradients are propagated in the same manner as the gradients in the standard backpropagation algorithm. The computation based on Equation 1 is fast and memory efficient because the procedure is similar to the standard backpropagation of gradients, which is done according to:

$$\nabla_{\theta_k} \ell = \frac{\partial \boldsymbol{x}_k}{\partial \theta_k} \cdot \frac{\partial \boldsymbol{x}_{k+1}}{\partial \boldsymbol{x}_k} \cdot \ldots \cdot \frac{\partial \boldsymbol{x}_n}{\partial \boldsymbol{x}_{n-1}} \cdot \nabla_{\boldsymbol{x}_n} \ell \tag{2}$$

The version of our algorithm that consists of two iterations (Algorithm 1) first performs the standard backpropagation (Equation 2) through layer outputs $\boldsymbol{x}$, and model parameters $\theta$ along with parameter update of an optimizer (in the experiments it is RMSProp) to new weight values $\theta'$. Then it is assumed that the absolute value of the parameter delta $|\theta - \theta'|$ of the RMSProp optimizer is good enough to retain it. The second backpropagation is performed for eventual negations of update directions only, where, conversely, the *average* gradient is propagated (Algorithm 2). Importantly, the range on which the gradient is averaged equals $[\theta, \theta']$ (between parameters before and after the estimated potential update; Algorithm 3). The average derivatives of each nonlinear activation are calculated as follows:

$$\underset{t \in [x,x']}{\mathcal{AVG}} f'(t) = \frac{\int_x^{x'} f'(t) \mathrm{d}t}{x' - x} = \frac{f(x') - f(x)}{x' - x} \tag{3}$$

where $f$ means an activation function (in the experiments it is either ELU or Tanh activation), $x$ means an input scalar assuming forward propagation using the $\theta$ weights, and $x'$ means the corresponding input number assuming forward pass for the $\theta'$. The equation is the one-dimensional analogy of the average gradient and the Jacobian, both of which are defined in Appendix A.

In the case of applying an activation function $f : \mathbb{R} \to \mathbb{R}$, or $\boldsymbol{f} : \mathbb{R}^n \to \mathbb{R}^n$, to a layer output $\boldsymbol{x} = \langle x_1, x_2, \ldots, x_n \rangle$ (assuming parameters $\theta$), which changes to $\boldsymbol{x}' = \langle x_1', x_2', \ldots, x_n' \rangle$ during the forward pass with updated parameters $\theta'$:

$$\underset{\boldsymbol{t} \in [\boldsymbol{x}, \boldsymbol{x}']}{\mathcal{AVG}} \frac{\partial \boldsymbol{f}}{\partial \boldsymbol{t}} = \mathrm{diag}(\langle \underset{t_1 \in [x_1, x_1']}{\mathcal{AVG}} f'(t_1), \underset{t_2 \in [x_2, x_2']}{\mathcal{AVG}} f'(t_2), \ldots, \underset{t_n \in [x_n, x_n']}{\mathcal{AVG}} f'(t_n) \rangle) \tag{4}$$

where each term $\mathcal{AVG}_{(\cdot)} f'(\cdot)$ is defined in Equation 3.

Let us define a typical layer, denoted as $k$, which is parameterized by $\theta_k$. This layer could be a convolutional layer, a fully-connected layer, or another operator that is linear over all or most of

its domain. Let us assume that the layer no. $k$ outputs $\boldsymbol{y}_k$, which is then passed to an activation $f_k$. Consequently each part of Equation 1 can be approximated as:

$$\mathcal{AVG}_{\boldsymbol{x}_k} \frac{\partial \boldsymbol{x}_{k+1}}{\partial \boldsymbol{x}_k} = \mathcal{AVG}_{\boldsymbol{x}_k} \frac{\partial f_k}{\partial \boldsymbol{x}_k} \cong \mathcal{AVG}_{\boldsymbol{x}_k} \frac{\partial \boldsymbol{y}_k}{\partial \boldsymbol{x}_k} \cdot \mathcal{AVG}_{\boldsymbol{t} \in [\boldsymbol{y}_k, \boldsymbol{y}_k']} \frac{\partial f}{\partial \boldsymbol{t}}$$
$$\mathcal{AVG}_{\theta_k} \frac{\partial \boldsymbol{x}_{k+1}}{\partial \theta_k} = \mathcal{AVG}_{\theta_k} \frac{\partial f_k}{\partial \theta_k} \cong \mathcal{AVG}_{\theta_k} \frac{\partial \boldsymbol{y}_k}{\partial \theta_k} \cdot \mathcal{AVG}_{\boldsymbol{t} \in [\boldsymbol{y}_k, \boldsymbol{y}_k']} \frac{\partial f}{\partial \boldsymbol{t}} \tag{5}$$

where the approximation, instead of equality, is the consequence of chaining averages of Jacobians, which can be proven analogously to Equation 1 (see Appendix B). The average operator $\mathcal{AVG}$ of Jacobians is defined in Appendix A. $\mathcal{AVG}_{\boldsymbol{t} \in [\boldsymbol{y}_{k+1}, \boldsymbol{y}_{k+1}']} \frac{\partial f}{\partial \boldsymbol{t}}$ is defined in Equation 4. Generally, the vast majority of applied neural network operators are either nonlinear activations or linear functions in by far most of their domains (e.g., max pooling, convolution, fully connected, or ReLU). In the case of the nonlinear activations, equations no. 3 and 4 are used to compute the average Jacobians. For linear transformations, such as $\boldsymbol{y}_k(\boldsymbol{x}_k)$ and $\boldsymbol{y}_k(\theta_k)$, the average gradients and Jacobians are easy and fast to compute. However, for implementation simplicity and a slight speedup of computations, broader estimates of the average Jacobians from Equation 5 are applied:

$$\mathcal{AVG}_{\boldsymbol{x}_k} \frac{\partial \boldsymbol{x}_{k+1}}{\partial \boldsymbol{x}_k} = \mathcal{AVG}_{\boldsymbol{x}_k} \frac{\partial f_k}{\partial \boldsymbol{x}_k} \cong \frac{\partial \boldsymbol{y}_k}{\partial \boldsymbol{x}_k} \cdot \mathcal{AVG}_{\boldsymbol{t} \in [\boldsymbol{y}_k, \boldsymbol{y}_k']} \frac{\partial f}{\partial \boldsymbol{t}}$$
$$\mathcal{AVG}_{\theta_k} \frac{\partial \boldsymbol{x}_{k+1}}{\partial \theta_k} = \mathcal{AVG}_{\theta_k} \frac{\partial f_k}{\partial \theta_k} \cong \frac{\partial \boldsymbol{y}_k}{\partial \theta_k} \cdot \mathcal{AVG}_{\boldsymbol{t} \in [\boldsymbol{y}_k, \boldsymbol{y}_k']} \frac{\partial f}{\partial \boldsymbol{t}} \tag{6}$$

which use the non-averaged Jacobian $\frac{\partial \boldsymbol{y}_k}{\partial \boldsymbol{x}_k}$. Therefore, intuitively, the broad estimation of $\mathcal{AVG}_{\boldsymbol{x}_k} \frac{\partial \boldsymbol{x}_{k+1}}{\partial \boldsymbol{x}_k}$ is approximately between $\frac{\partial \boldsymbol{x}_{k+1}}{\partial \boldsymbol{x}_k}$ and $\mathcal{AVG}_{\boldsymbol{x}_k} \frac{\partial \boldsymbol{x}_{k+1}}{\partial \boldsymbol{x}_k}$, and analogously for $\mathcal{AVG}_{\theta_k} \frac{\partial \boldsymbol{x}_{k+1}}{\partial \theta_k}$.

---

**Algorithm 1** *Simplified algorithm version for 2 iterations.* Back and forward propagation would be called two times in optimal implementation, where memory requirement would be the same as for Adam optimizer. Over the whole paper, we refer to the optimal implementation as the one that minimizes recomputations, avoids costly statistics during training, and is machine-code optimized to the same extent as optimizers from mainstream libraries. The lines marked as redundant within comments in curly brackets are unnecessary for the optimal operation of the below pseudocode.

**Input:** $model$: Neural Network Model
   $dataset$: Training Dataset
   $lossFn$: Loss Function
   $optimizer$: Optimizer
**for all** $batch \in dataset$ **do**
 $modelOutput \leftarrow model(batch.\text{x})$ {It is assumed that $model$'s layers' results are kept inside $model$}
 $modelLoss \leftarrow LossFn(modelOutput, batch.\text{y})$
 $modelCopy \leftarrow model$ {Copy $model$}
 $modelCopyOutput \leftarrow modelCopy(batch.\text{x})$ {This inference is redundant if $modelCopy$ gets also intermediate layers' results copied}
 $modelCopyLoss \leftarrow LossFn(modelCopyOutput, batch.\text{y})$ {This computation is also redundant, since it is the same as $modelLoss$}
 $Backpropagate(modelCopyLoss)$ {Compute the gradients using the standard backpropagation procedure. Assume that the gradients are stored inside $modelCopy$}
 $optimizer.\text{Step}(modelCopy)$ {Perform weight update on $modelCopy$ (using the gradients stored inside $modelCopy$)}
 $modelCopy(batch.\text{x})$ {Execute inference to store new layer-wise results in $modelCopy$}
 **AveragedBackpropagation**$(model, modelCopy, modelLoss)$ {The procedure is described as Algorithm 2. The parameters of the $model$ are modified within}
**end for**

---

An algorithm version with $n$ backpropagation iterations computes $(n-1)$ times the approximated gradient average, each time based on the previous. The intuition behind this is that a better estimate of the averaged derivatives of nonlinear activations is computed after every iteration (Equation 3; backpropagated according to equations no. 4, 6 and 1). Consequently, each time a more precise estimate of the optimal parameter update $(\Delta\theta)^*$ is obtained (where optimality means that the average gradient is accurately estimated and the parameter update is compliant with it). Therefore, once again

---

**Algorithm 2** *Averaged Backpropagation Algorithm* calculates the approximation of the average gradient.

---

**procedure** AVERAGEDBACKPROPAGATION(
$model$: Neural Network Model
$modelAfterUpdate$: $model$ After Candidate Update of
  Parameters
$modelLoss$: $model$'s Loss)
 $Backpropagate(modelLoss,$
  $model$.Layers.Last().Output) {Compute the gradient of $modelLoss$ in terms of the last
 layer's output. Let us assume that the gradient is assigned to the $grad$ property of the output
 variable ($model$.Layers.Last().Output)}
 **for** $index \leftarrow (Count(model.\text{Layers}) - 1)$ **to** 0 **do**
  $\theta = model$.Layers[$index$].$\theta$ {To simplify notation}
  **if** $IsNonlinear(model.\text{Layers}[index])$ **then** {Calculation of either the gradient or its aver-
  age, which corresponds to the terms in Equation 6}
   **BackpropagateThroughNonlinearLayer**($model$.Layers[$index$].Output,
    $model$.Layers[$index$].Input, $modelAfterUpdate$.Layers[$index$].Output,
    $modelAfterUpdate$.Layers[$index$].Input) {Procedure described as Algorithm 3. Let
   us assume that activations are separate layers (like in equations no. 4 and 6)}
  **else**
   $Backpropagate(model.\text{Layers}[index].\text{Output}, model.\text{Layers}[index].\text{Input})$ {The typ-
   ical backpropagation procedure. It propagates the gradient through a layer. Let us assume
   that the gradient is assigned to the $grad$ property of the input variable}
   $\theta.averagedGrad \leftarrow \theta.grad$ {In this case, for a linear layer, the gradient is treated as its
   average (compare equations no. 6 and 5)}
  **end if**
  $\theta' = modelAfterUpdate$.Layers[$index$].$\theta$ {Notation simplification}
  $\theta \leftarrow \theta + |\theta' - \theta| \cdot sgn(\theta.\text{averagedGrad})$ {Update by the absolute value of $optimizer$'s
  update from Algorithm 1: $|\theta' - \theta|$, but in the direction of the approximated gradient average
  $sgn(\theta.\text{averagedGrad})$}
 **end for**
**end procedure**

---

the average gradient can be refined to more precisely match the parameter update $(\Delta\theta)^*$, and so on (in a loop).

The $n$-iteration version of the method is labeled as Algorithm 4 in Appendix G (for two iterations it is a little slower than Algorithm 1 due to additional model-state copies, moreover it is more complex). The optimal implementation of the $n$-iteration algorithm variant would be slightly more than $n$ times slower than the gradient-based RMSprop training, where $n$ is the number of iterations. There are exactly $n$ backward passes, optimally $n$ inferences, and some additional copy operations $C$ of a model (optimally $|C| \leq n + 1$, but Algorithm 4 in Appendix G is suboptimal in this respect). However, the copies are generally significantly faster than forward or backward passes, because it is just needed to copy blocks of data, that are not bigger than the memory used during forward or backward pass. Furthermore, creating the copies by saving results of weight updates directly into different memory addresses only slightly increases the execution time.

An interesting way of comparing the gradient-based RMSProp optimization with our algorithm is to examine the average loss deltas for all weight updates of both algorithms. The purpose of the evaluation approach is to validate the proof in Appendix B.3. However, such a comparison focuses on loss measurements for a *single batch* each time, making it an imperfect predictor of performance on the *whole dataset*. The first iteration of our method is the gradient-based RMSProp procedure, hence the change in loss for RMSProp $\Delta_{RMSProp}$ is known for both the same model parameters and data as in the case of the loss delta of our method. Therefore, the sum of loss differences after the updates of both approaches can be easily and measurably compared relatively to the sum of loss

---

**Algorithm 3** *Backpropagation Through Nonlinear Layer.* It is assumed that each input number influences a corresponding single output scalar. This is because, in the experiments, the only operators assumed to be nonlinear during backpropagation of the gradient average are certain activation functions: $\mathbb{R} \to \mathbb{R}$).

---

   **procedure** BACKPROPAGATETHROUGHNONLINEARLAYER($LayerOutput$: Tensor
$LayerInput$: Tensor
$LayerOutputAfterUpdate$: Tensor
$LayerInputAfterUpdate$: Tensor)
     $f \leftarrow$ Layer function
     **for all** $(outputNum, inputNum, outputNumUpdated, inputNumUpdated) \in Zip($
       $LayerOutput, LayerInput, LayerOutputAfterUpdate, LayerInputAfterUpdate)$ **do**
     {The commonly used $Zip$ function illustrates iterating through multiple tensors at once}
       **if** $|inputNumUpdated - inputNum| > \epsilon$ **then** {Check if the difference in the inputs is
       higher than a tiny constant $\epsilon$. The condition prevents division by zero. In the experiments
       $\epsilon \approx 1.19\mathrm{e}{-7}$}
         $\mathcal{AVG}_{x \in [inputNum, inputNumUpdated]} \, f'(x) = \frac{outputNumUpdated - outputNum}{inputNumUpdated - inputNum}$ {Eq. 3 and
         4}
         $inputNum.\text{averagedGrad} \leftarrow \mathcal{AVG}_x \, f'(x) \cdot outputNum.\text{averagedGrad}$
         {Propagate the average gradient backward using the chain rule. Equations 3 and 4 define
         the term $\mathcal{AVG}_{t \in [y_k, y'_k]} \frac{\partial f}{\partial t}$ in Equation 6, which is part of Equation 1}
       **else**
         $inputNum.\text{averagedGrad} \leftarrow f'(inputNum) \cdot outputNum.\text{averagedGrad}$ {In this
         case $inputNum \approx inputNumUpdated$, thus $\mathcal{AVG}_{x \in [...]} \, f'(x) \approx f'(inputNum)$ for
         activation functions. The backpropagation towards input complies with the chain rule
         (equations no. 6 and 1)}
       **end if**
     **end for**
   **end procedure**

---

deltas of RMSProp:

$$\mathcal{RD}_{AG,RMSProp} = \frac{\mathcal{AVG}_{b \in B} \, \Delta_{AG} - \mathcal{AVG}_{b \in B} \, \Delta_{RMSProp}}{|\, \mathcal{AVG}_{b \in B} \, \Delta_{RMSProp}|}$$

$$= \frac{\sum_{b \in B} (\ell_b(\theta'_{AG,b}) - \ell_b(\theta_b))}{|\sum_{b \in B} (\ell_b(\theta'_{RMSProp,b}) - \ell_b(\theta_b))|} - sgn(\sum_{b \in B} (\ell_b(\theta'_{RMSProp,b}) - \ell_b(\theta_b))) \tag{7}$$

$\mathcal{RD}_{AG,RMSProp}$ is the relative difference in avg. loss deltas of $RMSProp$ and the method based on the average gradient ($AG$). The $\mathcal{AVG}$ operator denotes the arithmetic average. $B$ is the set of all batches. $\Delta_{AG,b}$ is the loss delta assuming a batch $b$ after our algorithm's update of model parameters $\theta_b$ to new values $\theta'_{AG,b}$. Notation for RMSProp is analogous. $\ell_b$ is the loss, assuming data of a batch $b$. $sgn$ is the sign function. $\mathcal{RD}$ would not be as useful when using momentum because the metric compares the aggregated loss of a single batch per parameter update, whereas momentum contributes to a decrease in loss over many batches per a single parameter update. Without the momentum, $\mathcal{RD}$ significantly increases the statistical confidence in comparing training algorithms because, for *the same* model weights, the losses are compared for each weight update. Keeping the same parameter values for each loss delta reduces the variance of $\mathcal{RD}$, resulting in a decrease in errors when comparing methods.

## 2.2 MODELS AND TRAINING

Our algorithm was tested on three different models with nonlinear ELU (Clevert et al., 2015) and Tanh activations. Model A has a small number of layers (Table 1), and the second one, Model B, is much deeper, with 30 nonlinear layers (Table 2; not counting max pooling as nonlinear). The third model is a convolutional neural network with 46 nonlinear layers (see Appendix H.1 for the training details of this model). It was assumed that Model A is trained for 15 epochs, while Model B – 500 in the case of the gradient-based RMSProp training, and 300 for our method. Grid search was used to find the optimal learning rates for the standard RMSProp training over the course of all 500 epochs,

while our method was optimized only for 200 epochs (out of 300 during testing). The objective of the hyperparameter search was to minimize the loss that is the smallest over a training. The results of the search for optimal learning rates are shown in Table 3. The epoch counts are tailored to ensure that the training achieves minimal or near-minimal test loss values before the final epoch of the gradient-based RMSProp training. The only loss function used in this research is cross-entropy loss, and the batch size is set to 128 in all experiments.

Table 1: *Model A*

| Layers | Output Shape | Parameter Count |
|---|---|---|
| Convolution 2D $(3 \times 3)$ + ELU | $(8, 26, 26)$ | 80 |
| Convolution 2D $(3 \times 3)$ + ELU | $(8, 24, 24)$ | 584 |
| Convolution 2D $(5 \times 5)$ + ELU stride $= 2$ padding $= 2$ | $(16, 12, 12)$ | 3216 |
| Convolution 2D $(3 \times 3)$ + ELU | $(16, 10, 10)$ | 2320 |
| Convolution 2D $(3 \times 3)$ + ELU | $(16, 8, 8)$ | 2320 |
| Convolution 2D $(5 \times 5)$ + ELU stride $= 2$ padding $= 2$ | $(16, 4, 4)$ | 6416 |
| Flatten | 256 | |
| Linear + Softmax | 10 | 2570 |
| | | 17506 |

Table 2: *Model B* is designed to test the performance of our algorithm on deep neural networks to achieve a reasonable time of many trainings for statistical significance of the results. The practicality of the architecture is not prioritized.

| Layers | Output Shape | Parameter Count |
|---|---|---|
| Convolution 2D $(3 \times 3)$ + ELU | $(8, 26, 26)$ | 80 |
| Max Pooling 2D $(2 \times 2)$ | $(8, 13, 13)$ | |
| Convolution 2D $(3 \times 3)$ + ELU | $(16, 11, 11)$ | 1168 |
| Max Pooling 2D $(2 \times 2)$ | $(16, 5, 5)$ | |
| Flatten | 400 | |
| Linear + Tanh | 10 | 4010 |
| **26**× Linear + Tanh | 10 | **26**×110 |
| Linear + Softmax | 10 | 330 |
| | | 8228 |

Models A and B were trained on two popular image datasets: MNIST (LeCun & Cortes, 2010) and Fashion MNIST (Xiao et al., 2017). Both datasets have the same input size $(28 \times 28 \times 1)$, but their image characteristics are *significantly* different. Moreover, since the method does not have any hyperparameters apart from the learning rate, it is less likely to overfit to a specific experimental setup (model, dataset, and learning rate) and show good results on it while experiencing deficient performance on other setups. We further validated the performance of our algorithm on a deep sequential convolutional model using an NLP benchmark, specifically the IMDB dataset. All details are presented in Appendix H.

## 3 RESULTS

For the shallow model A, all of the training algorithms are approximately equal (Fig. 2a, Fig. 2b). The relative difference in summed loss deltas (Equation 7) revealed that the algorithm based on the average gradient is only marginally better than the standard RMSprop according to $\mathcal{RD} = 1.20\mathrm{e}{-3} \pm 2.7\mathrm{e}{-4}$ (0.12% faster minimization of loss with 0.027% of SEM error) on MNIST and $\mathcal{RD} = 5.86\mathrm{e}{-3} \pm 2.79\mathrm{e}{-3}$ on Fashion MNIST in the case of two iterations. For five iterations, $\mathcal{RD} = 6.47\mathrm{e}{-4} \pm 9.8\mathrm{e}{-5}$ on MNIST and $\mathcal{RD} = 2.37\mathrm{e}{-3} \pm 4.5\mathrm{e}{-4}$ on Fashion MNIST.

The results of Model B are much more interesting. The version of the algorithm with two iterations is about three times faster at minimizing the median of training losses on both datasets (Fig. 2c; Fig. 2d). Moreover, the mean losses tend to be considerably lower than those for standard RMSProp training, even when repeating the experiments using different weight initialization (see Appendix I). Despite the minority of epochs with high oscillations, the method utilizing the average gradient is approximately two to three times faster in minimizing the mean loss, although this is not clearly visible in the plots. Furthermore, for both versions of our algorithm on both datasets, during from 49.3% to 70% of epochs, the average training loss was lower with statistical significance (SEM) than for the gradient-based RMSProp. Conversely, our algorithm was worse in that respect during from 0.667% to 2.33% of epochs with statistical significance. The average of minimal training losses on MNIST for the five iterations is $0.0393 \pm 0.0058$, which is significantly lower than $0.0883 \pm 0.0117$

Table 3: *Learning rates*. All hyperparameter searches of Model A consist of five trainings for each learning rate (LR), while in the case of Model B, it is one training, unless stated otherwise. For Model B, the losses do not directly predict the performance of the methods, because different epoch counts are used between the methods. The standard error of the mean is used as the confidence range for the losses, while for the LRs, the maximum distance to the next best LRs on both sides represents the errors. The LRs used in the experiments are listed in the "Learning Rate" column.

| Dataset | Model | Method | Learning Rate | The Most Important Hyperparameter Search Results [Learning Rate: Avg. of Min. Training Loss] |
|---|---|---|---|---|
| MNIST | Model A | RMSProp | **8e−4** | 6e−4 : 8.04e−3; 7e−4 : 6.48e−3; **8e−4 : 5.69e−3** 9e−4 : 5.94e−3; 10e−4 : 7.70e−3; 11e−4 : 7.39e−3 |
| | | 2 Iterations | **8e−4** | **8e−4 : 0.00555**; 9e−4 : 0.00692; 1e−3 : 0.00832 |
| | | 5 Iterations | **8e−4** | **8e−4 : 0.00514**; 9e−4 : 0.00580; 1e−3 : 0.00678 |
| | Model B | RMSProp | **2.5e−4** | 1.5e−4 : 0.194; 2e−4 : 0.0979; **2.5e−4 : 0.0651** 3e−4 : 0.0683; 3.5e−4 : 0.191; 4e−4 : 0.0759 The best learning rate of the search *after* the experiments (10 trainings per LR in $\{1.5e{-}4, 2e{-}4, \ldots, 5.5e{-}4\}$): $(3.5e{-}4 \pm 1.5e{-}4) : (0.0856 \pm 0.0139)$, (matches the performance in our experiments in Section 3) The loss for a high learning rate (10 trainings): $1.5e{-}3 : (2.09 \pm 0.05)$ |
| | | 2 Iterations | **7.5e−4** | The learning rate is guessed |
| | | 5 Iterations | **7.5e−4** | The learning rate is guessed |
| Fashion MNIST | Model A | RMSProp | **1.5e−3** | 1e−3 : 0.201; 1.25e−3 : 0.186; **1.5e−3 : 0.183** 1.75e−3 : 0.189; 2e−3 : 0.183; 2.25e−3 : 0.193 |
| | | 2 Iterations | **1.9e−3** | 1.8e−3 : 0.186; **1.9e−3 : 0.179**; 2e−3 : 0.180 |
| | | 5 Iterations | **1.5e−3** | **1.5e−3 : 0.178**; 1.6e−3 : 0.179; 1.7e−3 : 0.200 |
| | Model B | RMSProp | **3e−4** | 2e−4 : 0.356; 2.5e−4 : 0.331; **3e−4 : 0.285** 3.5e−4 : 0.349; 4e−4 : 0.487; 4.5e−4 : 0.459 The best learning rate of the search *after* the experiments (10 trainings per LR in $\{2e{-}4, 2.5e{-}4, \ldots, 6e{-}4\}$): $(4e{-}4 \pm 1.5e{-}4) : (0.318 \pm 0.016)$, (matches the performance in our experiments in Section 3) The loss for a high learning rate (10 trainings): $9e{-}4 : (0.641 \pm 0.168)$ |
| | | 2 Iterations | **9e−4** | 6e−4 : 0.330; **9e−4 : 0.242**; 1.2e−3 : 0.355 |
| | | 5 Iterations | **9e−4** | **9e−4 : 0.243**; 1.5e−3 : 0.276 |

for the standard RMSProp. Meanwhile, the two iterations are also perform better than the gradient-based RMSProp, but without statistical significance, achieving $0.0747 \pm 0.0188$. Even better averages of minimal training losses were obtained on Fashion MNIST, with the five-iteration and two-iteration versions achieving $0.254 \pm 0.017$ and $0.257 \pm 0.014$ respectively, compared to $0.314 \pm 0.008$ by the gradient-based training.

Plots of the test losses of Model B look very similar to the training losses (Appendix C), showing significant improvements in generalization, which correspond to the lower training losses. On MNIST, the average of best accuracies over training for five iterations is equal to $(97.87 \pm 0.09)\%$, which is significantly higher than $(96.80 \pm 0.78)\%$ and $(96.75 \pm 0.55)\%$ for the two-iteration version and gradient-based algorithm, respectively. On Fashion MNIST, the analogous results are $(88.09 \pm 0.35)\%$, $(87.54 \pm 0.55)\%$ and $(86.57 \pm 0.29)\%$, respectively. Appendix D presents the accuracy plots.

For Model B, the $\mathcal{RD}$ metric (Equation 7) provides a very high confidence of superiority of the average gradient for the high learning rates used for the trainings based on our algorithm (Table 3). On MNIST for two and five iterations, it equals $10.41 \pm 1.94$ and $1.43 \pm 0.29$, respectively. On Fashion MNIST, it is $0.58 \pm 0.14$ and $0.24 \pm 0.04$ for both variants, respectively.

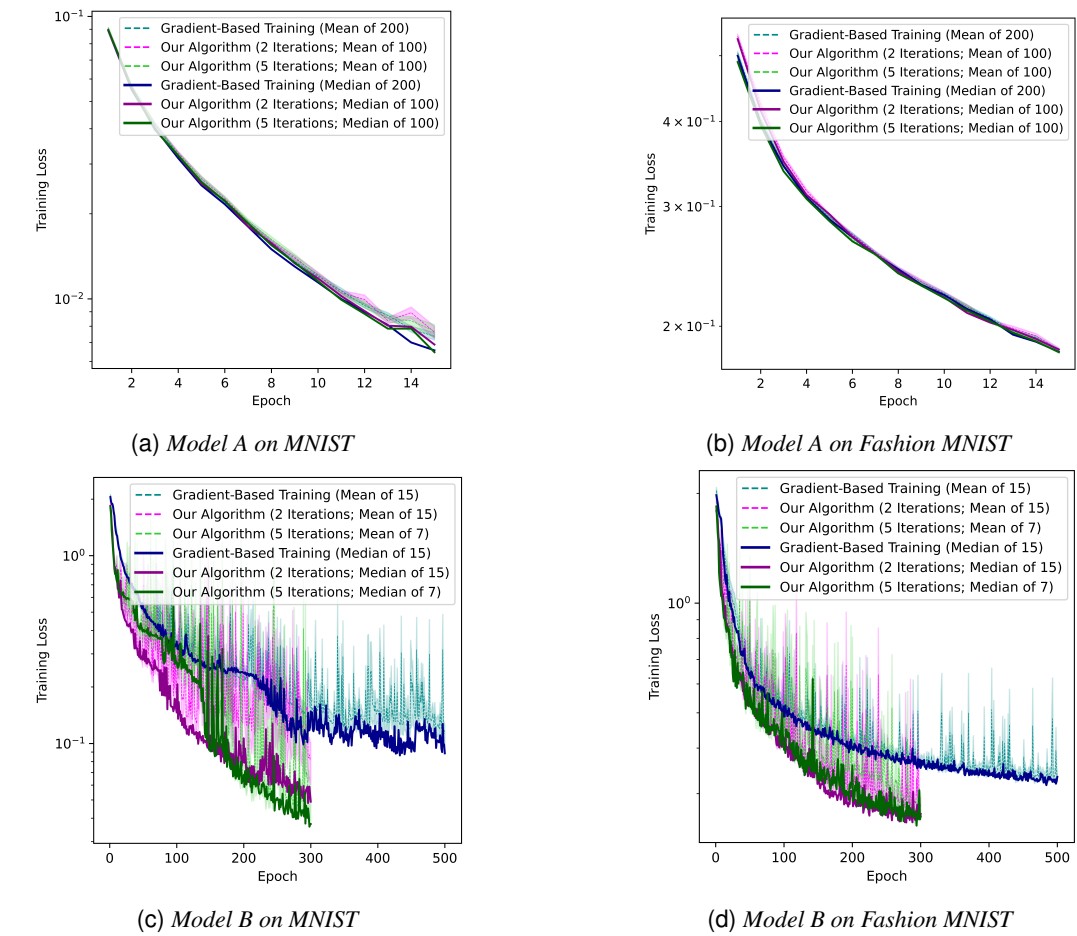

(a) *Model A on MNIST*

(b) *Model A on Fashion MNIST*

(c) *Model B on MNIST*

(d) *Model B on Fashion MNIST*

Figure 2: *Training losses*. Only mean curves contain confidence ranges (SEM).

Importantly, for Model B, the average-gradient algorithm dominated also for the learning rates that are optimal for the standard RMSProp training. Multiple metrics favored our algorithm with statistical significance , i.e., $\mathcal{RD} \in [0.0611 \pm 0.0004, 1.07 \pm 0.31]$, despite training counts equal to only two or three (for each of the four experiments).

In the case of Model B, our implementation of the two-iteration variant of the algorithm based on the average gradient (Alg. 1) is nearly three times slower per epoch than the training based on the gradient, while the five iterations (Alg. 4 in Appendix G) are almost eight times slower per epoch. The estimated runtime of optimal implementation is slightly more than two times longer for the two iterations per epoch when compared to the gradient-based RMSProp, and around six to seven times longer for the five iterations.

For the deep convolutional model on the IMDB dataset, sample efficiency of the two iteration variant of our algorithm achieved about 55% gain in sample efficiency compared to the gradient baseline. The analogous result using four iterations falls between 25% and 30%. See Appendix H.2 for details.

## 4 Conclusions

Surprisingly, modifying the gradient on nonlinear activations in very deep models can significantly increase sample efficiency for some deep models, which is a direct conclusion of our experiments. For the MNIST and Fashion MNIST benchmarks, the algorithm based on the average gradient offers significant benefits compared to the standard RMSProp training for the deep model with many stacked fully-connected layers and nonlinear activations: (a) About a threefold increase in sample efficiency

in terms of median loss, and about two to three times faster mean loss reduction. This is reached by only two iterations, which optimally require a little more than double the time of computation per epoch in comparison with the gradient-based RMSProp training. Meanwhile, our suboptimal implementation of the two-iteration version of the algorithm needs nearly three times more runtime per epoch than the training based on the gradient. Therefore, the presented method is not only more sample-efficient, but it is also faster and saves energy. (b) Outstanding performance on higher learning rates, which may offer significant benefits in terms of both electricity and time spent on hyperparameter searches. (c) Considerably better generalization, at least in a reasonable epoch count. The increase in sample efficiency and good performance across a wider range of learning rates is confirmed by experiments using different weight initialization (see Appendix I).

On the other hand, for a deep sequential convolutional model trained on the IMDB dataset, sample efficiency is improved by about 55% when using only two iterations of our algorithm (Appendix H.2). This is the only significant benefit of our algorithm in this experiment, as the variant using more iterations achieved efficiency between that of the vanilla RMSProp and the two-iteration variant.

The $\mathcal{RD}$ (Equation 7) confirms the outstanding results of the other measures. The score of $\mathcal{RD} = 10.41 \pm 1.94$, achieved by the two iterations on MNIST, corresponds to the average speed of batch-loss minimization that is $(1141 \pm 194)\%$ of the speed of the gradient-based RMSProp while using the same absolute values of weight updates. In the other cases of deep models, the average speed of batch-loss minimization ranges from $(2.10 \pm 0.18)\%$ to $(243 \pm 29)\%$. Therefore, even a relatively slight speedup in batch-loss minimization (such as 2.1% on the IMDB dataset) can contribute to a significantly higher gain in sample efficiency. Moreover, it is crucial to note that the highest of the mentioned gains occur at learning rates that are three times higher than the optimal rates for gradient-based training. Generally, high learning rate values may enable rapid learning because model parameters are adjusted faster. Nevertheless, the average gradient is also superior in terms of the average speed of batch-loss minimization when using the optimal learning rates for gradient-based training across all tested models, with statistical significance. This validates the proof in Appendix B, as both the metric and the proof focus on the efficiency of batch-loss minimization. On the other hand, refer to Appendix F for the limitations of our algorithm in estimating the locally optimal update.

Surprisingly, the algorithm version with five iterations is worse than the two iterations according to $\mathcal{RD}$ with higher statistical confidence than for other measures. Across all experiments, the variant is computationally inefficient in terms of the resources required to reduce the loss to a certain level.

In the case of the shallow model with nonlinear ELU activations, the method is only marginally better than the standard gradient-based RMSProp training. This behavior is expected due to the scaling properties of the algorithm (Appendix E).

## 5 DISCUSSION

The successful evaluation using different weight initialization techniques on the NLP and computer vision benchmarks, using both deep convolutional architecture and the model based on fully-connected layers with nonlinear activations, provides insight into significant improvements in sample efficiency, at least for some models. Furthermore, the computational cost associated with these improvements is modest. These results are especially important in the field of online learning, where sample efficiency is crucial.

For very deep models without residual connections, gradient-based training tends to be inefficient (Balduzzi et al., 2017), which we demonstrate how to mitigate. In general, the very deep structure of human brains enables the learning of universal and complex patterns. Therefore, accurately mimicking human brain model could potentially lead to satisfactory results. Our algorithm aims to improve learning in scenarios involving neural structures that are very deep, a feature of provably efficient biological brains that distinguishes them from current AI models. Therefore, the method may contribute to the training of large models in the future, where sample efficiency is needed to learn new tasks on the fly, akin to how people or some animals do.

However, at present, the potential modifications to the algorithm are even more intriguing. Not only is it possible to efficiently calculate the average gradient for linear layers using Eq. 5 instead of Eq. 6, but Eq. 1 can also be utilized to compute the average gradients over a much larger range than that of a parameter update to capture the global trend of the loss landscape (see Appendix J for future work).

## 6 REPRODUCIBILITY STATEMENT

We put emphasis on providing detailed descriptions of all experiments. The algorithms (Alg. 4 in Appendix G and Alg. 1 in Section 2, with subprocedures labeled as Alg. 2 and Alg. 3) are described in detail in Section 2.1. The models (Tables 1, 2 and 4 in Appendix H), the learning rates (Tables 3 and 5 in Appendix H), and all other important experiment settings are described in Section 2.2 and Appendix H.1. The code, along with environment settings, is available under [...]. Appendix B contains one of our most important theoretical results: the proof of Equation 1 and its superiority over the gradient in minimizing the batch loss by accurately indicating how each model parameter individually contributes to the change in the batch loss (Equation 14). The proven potential for batch-loss minimization is verified not only by the $\mathcal{RD}$ metric with high statistical significance but also by comparisons of training losses and other metrics (Section 3).

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

## A    DEFINITION OF AVERAGE GRADIENT/JACOBIAN

Let us define the average gradient of a function $f(\boldsymbol{x}) : \mathbb{R}^n \to \mathbb{R}$ for some row vector $\boldsymbol{x} \in [\boldsymbol{a}, \boldsymbol{b}]$ (the formula is analogous to the one-dimensional case in Equation 3):

$$\underset{\boldsymbol{x}\in[\boldsymbol{a},\boldsymbol{b}]}{\mathcal{AVG}} \nabla_{\boldsymbol{x}} f = (\boldsymbol{b} - \boldsymbol{a})^{\circ-1} \circ \int_{\boldsymbol{a}}^{\boldsymbol{b}} \nabla_{\boldsymbol{x}} f \; d\boldsymbol{x} = (\boldsymbol{b} - \boldsymbol{a})^{\circ-1} \circ \int_0^1 \nabla_{\boldsymbol{a}+t\cdot(\boldsymbol{b}-\boldsymbol{a})} f \; dt \tag{8}$$

where $\circ$ denotes the elementwise operation of either multiplication or inversion $((\cdot)^{\circ-1})$. However, the cases of vector elements where division by zero occurs are handled differently, using the partial derivative $\frac{\partial f}{\partial x_i}$:

$$\forall i : b_i - a_i = 0 \implies \underset{x_i\in[a_i,b_i]}{\mathcal{AVG}} \frac{\partial f}{\partial x_i} = \frac{\partial f}{\partial a_i} \tag{9}$$

If $\boldsymbol{f}(\boldsymbol{x}) : \mathbb{R}^n \to \mathbb{R}^m$, then using to Equation 8:

$$
\underset{\boldsymbol{x} \in [\boldsymbol{a}, \boldsymbol{b}]}{\mathcal{AVG}} \frac{\partial \boldsymbol{f}}{\partial \boldsymbol{x}} = \begin{bmatrix} \mathcal{AVG}_{\boldsymbol{x} \in [\boldsymbol{a},\boldsymbol{b}]} \nabla_{\boldsymbol{x}} f_1 \\ \mathcal{AVG}_{\boldsymbol{x} \in [\boldsymbol{a},\boldsymbol{b}]} \nabla_{\boldsymbol{x}} f_2 \\ \cdots \\ \mathcal{AVG}_{\boldsymbol{x} \in [\boldsymbol{a},\boldsymbol{b}]} \nabla_{\boldsymbol{x}} f_m \end{bmatrix} = \begin{bmatrix} (\boldsymbol{b} - \boldsymbol{a})^{\circ -1} \circ \int_{\boldsymbol{a}}^{\boldsymbol{b}} \nabla_{\boldsymbol{x}} f_1 \ d\boldsymbol{x} \\ (\boldsymbol{b} - \boldsymbol{a})^{\circ -1} \circ \int_{\boldsymbol{a}}^{\boldsymbol{b}} \nabla_{\boldsymbol{x}} f_2 \ d\boldsymbol{x} \\ \cdots \\ (\boldsymbol{b} - \boldsymbol{a})^{\circ -1} \circ \int_{\boldsymbol{a}}^{\boldsymbol{b}} \nabla_{\boldsymbol{x}} f_m \ d\boldsymbol{x} \end{bmatrix}
$$
$$
= \begin{bmatrix} (\boldsymbol{b} - \boldsymbol{a})^{\circ -1} \\ (\boldsymbol{b} - \boldsymbol{a})^{\circ -1} \\ \cdots \\ (\boldsymbol{b} - \boldsymbol{a})^{\circ -1} \end{bmatrix} \circ \begin{bmatrix} \int_0^1 \nabla_{\boldsymbol{a} + t \cdot (\boldsymbol{b} - \boldsymbol{a})} f_1 \ dt \\ \int_0^1 \nabla_{\boldsymbol{a} + t \cdot (\boldsymbol{b} - \boldsymbol{a})} f_2 \ dt \\ \cdots \\ \int_0^1 \nabla_{\boldsymbol{a} + t \cdot (\boldsymbol{b} - \boldsymbol{a})} f_m \ dt \end{bmatrix} \tag{10}
$$

Again, the cases of vector elements where division by zero occurs are handled as follows:

$$
\forall i : b_i - a_i = 0 \implies \underset{x_i \in [a_i, b_i]}{\mathcal{AVG}} \frac{\partial \boldsymbol{f}}{\partial x_i} = \frac{\partial \boldsymbol{f}}{\partial a_i} \tag{11}
$$

# B  PROOF OF EQUATION 1 AND ITS LOSS-MINIMIZATION POTENTIAL

## B.1  DEFINITION AND PROPERTIES OF AVERAGE GRADIENT OF LOSS

Using Equation 2, the average gradient $\mathcal{AVG}_{\theta_k} \nabla_{\theta_k} \ell$ can be defined without the approximation given in Equation 1:

$$
\underset{\theta_k}{\mathcal{AVG}} \nabla_{\theta_k} \ell = \underset{(\theta_k, \boldsymbol{x}_k, \boldsymbol{x}_{k+1}, \dots, \boldsymbol{x}_n)}{\mathcal{AVG}} \left( \frac{\partial \boldsymbol{x}_k}{\partial \theta_k} \cdot \frac{\partial \boldsymbol{x}_{k+1}}{\partial \boldsymbol{x}_k} \cdot \dots \cdot \frac{\partial \boldsymbol{x}_n}{\partial \boldsymbol{x}_{n-1}} \cdot \nabla_{\boldsymbol{x}_n} \ell \right) \tag{12}
$$

where multiple variables are under the average operator $(\theta_k, \boldsymbol{x}_k, \boldsymbol{x}_{k+1}, \dots, \boldsymbol{x}_n)$. There are numerous ways to define how $(\boldsymbol{x}_k, \boldsymbol{x}_{k+1}, \dots, \boldsymbol{x}_n)$ depend on the weights and biases $\theta_k$, as they all change together during a parameter update. To compute the average (Equation 12), it can be assumed that the parameters of the layer no. $k$ and the outputs of the layers change linearly with respect to each other, as if they move from $\theta_k$ to $\theta_k'$ and from $(\boldsymbol{x}_k, \dots, \boldsymbol{x}_n)$ to $(\boldsymbol{x}_k', \dots, \boldsymbol{x}_n')$ after an update of the parameters of all layers. Under this assumption, the calculation is formulated as follows: while computing the average, the integral contains a function $f_{\theta,k}(t) = \theta_k + t \cdot (\theta_k' - \theta_k)$ for the variable under integration $t \in [0, 1]$ ($\theta_k$ and $\theta_k'$ denote model parameters before and after an update, respectively). Moreover, the integral involves each layer's output: $\boldsymbol{f}_{\boldsymbol{x},i}(t) = \boldsymbol{x}_i + t \cdot (\boldsymbol{x}_i' - \boldsymbol{x}_i)$. Finally, the average gradient (Equation 12) is equal to:

$$
\underset{\theta_k}{\mathcal{AVG}} \nabla_{\theta_k} \ell = \underset{f_{\theta,k}}{\mathcal{AVG}} \nabla_{f_{\theta,k}} \ell = \underset{t}{\mathcal{AVG}} \left( \frac{\partial \boldsymbol{f}_{\boldsymbol{x},k}(t)}{\partial f_{\theta,k}(t)} \cdot \frac{\partial \boldsymbol{f}_{\boldsymbol{x},k+1}(t)}{\partial \boldsymbol{f}_{\boldsymbol{x},k}(t)} \cdot \dots \cdot \frac{\partial \boldsymbol{f}_{\boldsymbol{x},n}(t)}{\partial \boldsymbol{f}_{\boldsymbol{x},n-1}(t)} \cdot \nabla_{\boldsymbol{f}_{\boldsymbol{x},n}(t)} \ell(t) \right)
$$
$$
= \int_0^1 \frac{\partial \boldsymbol{f}_{\boldsymbol{x},k}(t)}{\partial f_{\theta,k}(t)} \cdot \frac{\partial \boldsymbol{f}_{\boldsymbol{x},k+1}(t)}{\partial \boldsymbol{f}_{\boldsymbol{x},k}(t)} \cdot \dots \cdot \frac{\partial \boldsymbol{f}_{\boldsymbol{x},n}(t)}{\partial \boldsymbol{f}_{\boldsymbol{x},n-1}(t)} \cdot \nabla_{\boldsymbol{f}_{\boldsymbol{x},n}(t)} \ell(t) \ dt \tag{13}
$$

which is more direct and easier to work with.

Importantly, unlike the gradient, the average gradient $(\mathcal{AVG}_{\theta_k} \nabla_{\theta_k} \ell)$ is directly proportional to the loss-change impact of each model parameter separately $\boldsymbol{l}_{\theta',k} - \boldsymbol{l}_{\theta,k}$ (of the shape of $\theta_k$ and $\theta_k'$, unlike the scalar $\ell$):

$$
\underset{\theta_k}{\mathcal{AVG}} \nabla_{\theta_k} \ell = \underset{\theta_k}{\mathcal{AVG}} \nabla_{\theta_k} \left( \sum_{j=0}^n \sum_{i=0}^{|\theta_j|} \ell_{\theta,j,i} \right) = \underset{\theta_k}{\mathcal{AVG}} \nabla_{\theta_k} \left( \sum_{i=0}^{|\theta_k|} \ell_{\theta,k,i} \right) = \underset{\theta_k}{\mathcal{AVG}} \left( \text{diag}\left( \frac{\partial \boldsymbol{l}_{\theta,k}}{\partial \theta_k} \right) \right) =
$$
$$
= \langle \underset{\theta_{k,1}}{\mathcal{AVG}} \ell_{\theta,k,1}', \dots, \underset{\theta_{k,n}}{\mathcal{AVG}} \ell_{\theta,k,n}' \rangle = \langle \frac{\int_{\theta_{k,1}}^{\theta_{k,1}'} \ell_{\vartheta,k,1}' \ d\vartheta}{\theta_{k,1}' - \theta_{k,1}}, \dots, \frac{\int_{\theta_{k,n}}^{\theta_{k,n}'} \ell_{\vartheta,k,n}' \ d\vartheta}{\theta_{k,n}' - \theta_{k,n}} \rangle \tag{14}
$$
$$
= \langle \frac{\ell_{\theta',k,1} - \ell_{\theta,k,1}}{\theta_{k,1}' - \theta_{k,1}}, \dots, \frac{\ell_{\theta',k,n} - \ell_{\theta,k,n}}{\theta_{k,n}' - \theta_{k,n}} \rangle = (\theta_k' - \theta_k)^{\circ -1} \circ (\boldsymbol{l}_{\theta',k} - \boldsymbol{l}_{\theta,k}) \propto \boldsymbol{l}_{\theta',k} - \boldsymbol{l}_{\theta,k}
$$

where $\circ$ denotes the elementwise operation of either multiplication or inversion $((\cdot)^{\circ-1})$. $\mathrm{diag}(\frac{\partial \boldsymbol{l}_{\theta,k}}{\partial \theta_k})$ denotes diagonal elements of the Jacobian matrix. $\ell_{\theta,k,i} \in \boldsymbol{l}_{\theta,k}$ represents the scalar loss contribution of a single model parameter ($\theta_{k,i}$), that can be defined as an integral of the gradient: $\ell_{\theta,k,i} = \int_{C_1}^{\theta_{k,i}} \nabla_{\vartheta} \ell_{\theta} \, d\vartheta + C_2$, for any constant scalars $C_1$ and $C_2$. (Note that in this case, $\ell_{\theta,k,i} \neq \ell_{\theta} + C_{\ell}$, for any constant $C_{\ell}$, because the loss $\ell$ also depends on other parameters than $\theta_{k,i}$.) Important properties: (a) $\ell_{\theta} = C + \sum_{k=0}^{n} \sum_{i=0}^{|\theta_k|} \ell_{\theta,k,i}$ for a constant $C$ that is invariant across updates of the model parameters $\theta$. (b) The elements of $\boldsymbol{l}$ are related to the difference in loss during parameter update: $\ell_{\theta'} - \ell_{\theta} = (\sum_{k=0}^{n} \sum_{i=0}^{|\theta'_k|} \ell_{\theta',k,i}) - (\sum_{k=0}^{n} \sum_{i=0}^{|\theta_k|} \ell_{\theta,k,i})$. (c) The following equation is satisfied: $\nabla_{\theta} \ell = \nabla_{\theta} (\sum_{k=0}^{n} \sum_{i=0}^{|\theta_k|} \ell_{\theta,k,i})$. The simple one-dimensional visualization of the proportionality from Equation 14 ($\mathcal{AVG}_{\theta_k} \nabla_{\theta_k} \ell \propto \boldsymbol{l}_{\theta',k} - \boldsymbol{l}_{\theta,k}$) is shown in Fig. 1. Note that the property of proportionality does not hold for the gradient updates (which are utilized by Adam (Kingma & Ba, 2014), RMSProp (Tieleman et al., 2012), and SGD (Ketkar, 2017; Liu et al., 2020)). In the gradient case, during the update step of $\theta$ weights, $\theta'$ is not used in the calculation of itself. Therefore, $\boldsymbol{l}_{\theta'} - \boldsymbol{l}_{\theta}$ cannot be computed yet, and the accurate influence on loss remains unknown, unlike for the average gradient (Equation 14). The cases of scalar parameters $\theta_{k,i} \in \theta_k$ and $\theta'_{k,i} \in \theta'_k$ where division by zero occurs are handled differently:

$$\forall i: \theta'_{k,i} - \theta_{k,i} = 0 \implies \underset{\vartheta_{k,i} \in [\theta_{k,i}, \theta'_{k,i}]}{\mathcal{AVG}} \frac{\partial \ell}{\partial \vartheta_{k,i}} = \frac{\partial \ell}{\partial \theta_{k,i}} \tag{15}$$

Assuming the functions $f_{\theta,k}$ and $\boldsymbol{f}_{\boldsymbol{x},i}$ from Equation 13 are any functions (but differentiable with respect to each other), Equation 14 remains valid. Therefore, the crucial property of direct proportionality to the loss values does not depend on our previous assumptions about $\theta_k$ and $\boldsymbol{x}_i$. The purpose of these assumptions is to provide a simple example, reduce reasoning abstraction, and simplify further proofs in Sections B.2 and B.3.

## B.2 PROOF OF OF EQUATION 1 WITHOUT SPECIFYING PRECISION OF APPROXIMATION

For some function $f$ and some constants $C_1, C_2, \ldots, C_n$:

$$\int C_1 \cdot C_2 \cdot \ldots \cdot C_n \cdot f(x) \, dx = C_1 \cdot C_2 \cdot \ldots \cdot C_n \cdot \int f(x) \, dx \tag{16}$$

Similarly, let us denote approximately constant functions as $C'_1(x) \cong C_1, C'_2(x) \cong C_2, \ldots, C'_n(x) \cong C_n$ for some $x \in [a, b]$, $a \neq b$. The constant that precisely approximates each function $C''(x)$, is its average: $C'_1(x) \cong \mathcal{AVG}\, C'_1(x) = C_1, C'_2(x) \cong \mathcal{AVG}\, C'_2(x) = C_2, \ldots, C'_n(x) \cong \mathcal{AVG}\, C'_n(x) = C_n$. Therefore, similarly to Equation 16:

$$\int_a^b C'_1(x) \cdot \ldots \cdot C'_n(x) \cdot f(x) \, dx \cong \underset{x \in [a,b]}{\mathcal{AVG}}\, C'_1(x) \cdot \ldots \cdot \underset{x \in [a,b]}{\mathcal{AVG}}\, C'_n(x) \cdot \int_a^b f(x) \, dx$$
$$\int_a^b C'_1(x) \cdot \ldots \cdot C'_n(x) \cdot f(x) \, dx \cong \int_a^b \frac{C'_1(x)}{b-a} dx \cdot \ldots \cdot \int_a^b \frac{C'_n(x)}{b-a} dx \cdot \int_a^b f(x) \, dx \tag{17}$$

which is also approximately equal to both sides of Equation 16. In Equation 17, both approximations are equivalent, because $\mathcal{AVG}\, C'_i(x) = \int_a^b C'_i(x)/(b-a)\, dx$. For functions $\mathbb{R}^n \to \mathbb{R}^m$, equations no. 16 and 17 are analogous. Note that, in the general case, the different approximations of the terms $C'_i(x) \cong C'_i(a)$ and $C'_i(x) \cong C_i(b)$ are worse than the average: $C'_i(x) \cong \mathcal{AVG}\, C'_i(x) = C_i$ (which is used further in Section B.3).

Rapid changes in the gradient over the range of an update indicate that the update step is too large, leading to instability and reduced training effectiveness due to excessively large steps in the loss landscape. We assume effective learning, where gradients do not change significantly[1] between updates, ensuring the learning rate is appropriately sized. In this case, the gradient $\nabla_{\theta_k} \ell$ does not change significantly[2] over the range of a weight update $[\theta, \theta']$. However, these assumptions are

---

[1]The magnitude of the gradient change need not be specified, as it suffices that it contributes to the approximations with unspecified bounds in Equations 18 and 19. The accuracy of these approximations is proven in Section B.3.

[2]See footnote 1.

merely intended to build intuition and *are not necessary for this proof.* We do not yet assume any specific level of precision in how Equation 17 approximates Equation 13:

$$
\begin{aligned}
\mathcal{AVG}_{\theta_k} \nabla_{\theta_k} \ell &= \int_0^1 \frac{\partial \boldsymbol{f}_{\boldsymbol{x},k}(t)}{\partial f_{\theta,k}(t)} \cdot \frac{\partial \boldsymbol{f}_{\boldsymbol{x},k+1}(t)}{\partial \boldsymbol{f}_{\boldsymbol{x},k}(t)} \cdot \ldots \cdot \frac{\partial \boldsymbol{f}_{\boldsymbol{x},n}(t)}{\partial \boldsymbol{f}_{\boldsymbol{x},n-1}(t)} \cdot \nabla_{\boldsymbol{f}_{\boldsymbol{x},n}(t)} \ell(t) \, dt \\
&\cong \int_0^1 \frac{\partial \boldsymbol{f}_{\boldsymbol{x},k}(t)}{\partial f_{\theta,k}(t)} \, dt \cdot \int_0^1 \frac{\partial \boldsymbol{f}_{\boldsymbol{x},k+1}(t)}{\partial \boldsymbol{f}_{\boldsymbol{x},k}(t)} \, dt \cdot \ldots \cdot \int_0^1 \frac{\partial \boldsymbol{f}_{\boldsymbol{x},n}(t)}{\partial \boldsymbol{f}_{\boldsymbol{x},n-1}(t)} \, dt \cdot \int_0^1 \nabla_{\boldsymbol{f}_{\boldsymbol{x},n}(t)} \ell(t) \, dt \quad (18) \\
&= \mathcal{AVG}_{\theta_k} \frac{\partial \boldsymbol{x}_k}{\partial \theta_k} \cdot \mathcal{AVG}_{\boldsymbol{x}_k} \frac{\partial \boldsymbol{x}_{k+1}}{\partial \boldsymbol{x}_k} \cdot \ldots \cdot \mathcal{AVG}_{\boldsymbol{x}_{n-1}} \frac{\partial \boldsymbol{x}_n}{\partial \boldsymbol{x}_{n-1}} \cdot \mathcal{AVG}_{\boldsymbol{x}_n} \nabla_{\boldsymbol{x}_n} \ell
\end{aligned}
$$

Applying the notation of Equation 12 to Equation 18, we get:

$$
\begin{aligned}
\mathcal{AVG}_{\theta_k} \nabla_{\theta_k} \ell &\cong \int_0^1 \frac{\partial \boldsymbol{f}_{\boldsymbol{x},k}(t)}{\partial f_{\theta,k}(t)} \, dt \int_0^1 \frac{\partial \boldsymbol{f}_{\boldsymbol{x},k+1}(t)}{\partial \boldsymbol{f}_{\boldsymbol{x},k}(t)} \, dt \cdot \ldots \cdot \int_0^1 \frac{\partial \boldsymbol{f}_{\boldsymbol{x},n}(t)}{\partial \boldsymbol{f}_{\boldsymbol{x},n-1}(t)} \, dt \int_0^1 \nabla_{\boldsymbol{f}_{\boldsymbol{x},n}(t)} \ell(t) \, dt \\
&= \int_{\theta_k}^{\theta_k'} \frac{\partial \boldsymbol{x}_k(\vartheta_k)}{\partial \vartheta_k} \, d\vartheta_k \int_{\boldsymbol{x}_k'}^{\boldsymbol{x}_k} \frac{\partial \boldsymbol{x}_{k+1}(\boldsymbol{\chi}_k)}{\partial \boldsymbol{\chi}_k} \, d\boldsymbol{\chi}_k \cdot \ldots \cdot \int_{\boldsymbol{x}_{n-1}}^{\boldsymbol{x}_{n-1}'} \frac{\partial \boldsymbol{x}_n(\boldsymbol{\chi}_{n-1})}{\partial \boldsymbol{\chi}_{n-1}} \, d\boldsymbol{\chi}_{n-1} \int_{\boldsymbol{x}_n}^{\boldsymbol{x}_n'} \nabla_{\boldsymbol{\chi}_n} \ell \, d\boldsymbol{\chi}_n \\
&= \mathcal{AVG}_{\theta_k} \frac{\partial \boldsymbol{x}_k}{\partial \theta_k} \cdot \mathcal{AVG}_{\boldsymbol{x}_k} \frac{\partial \boldsymbol{x}_{k+1}}{\partial \boldsymbol{x}_k} \cdot \ldots \cdot \mathcal{AVG}_{\boldsymbol{x}_{n-1}} \frac{\partial \boldsymbol{x}_n}{\partial \boldsymbol{x}_{n-1}} \cdot \mathcal{AVG}_{\boldsymbol{x}_n} \nabla_{\boldsymbol{x}_n} \ell
\end{aligned}
$$

$$(19)$$

where $\theta_k, \boldsymbol{x}_k, \boldsymbol{x}_{k+1}, \ldots, \boldsymbol{x}_n$ are all linear functions of $t$ (previously denoted as $f_{\theta,k}, f_{\boldsymbol{x},k}, f_{\boldsymbol{x},k+1}, \ldots, f_{\boldsymbol{x},n}$). Therefore, the functions $\boldsymbol{x}_k(\theta_k), \boldsymbol{x}_{k+1}(\boldsymbol{x}_k), \ldots, \boldsymbol{x}_n(\boldsymbol{x}_{n-1})$ are known. The edge cases of those scalars within $\theta_k, \boldsymbol{x}_k, \boldsymbol{x}_{k+1}, \ldots, \boldsymbol{x}_n$ that do not depend on $t$ are handled analogously to Equation 15, as in these cases the average gradient equals the gradient.

Despite the provided arguments on why the approximation is applied, the precision of the estimation is not specified, although it *is crucial.* Therefore, the accuracy of the approximation is described in Section B.3. Otherwise, if the precision of the estimation is not important, then Equation 19 ultimately proves Equation 1. $\square$

The analogous reasoning can be applied to prove Equation 5.

In the algorithm, it is also assumed that the average gradient of the loss with respect to the output of the last layer, denoted as $(\mathcal{AVG}_{\boldsymbol{x}_n} \nabla_{\boldsymbol{x}_n} \ell)$, is replaced by the gradient $(\nabla_{\boldsymbol{x}_n} \ell)$. Moreover, in our implementation, the gradients replace the average gradients of layers that are approximately linear (using Equation 6 instead of Equation 5), resulting in a broader approximation in Equation 1. However, the presented reasoning still applies, including the proof of approximation accuracy in Section B.3. See Appendix F for comments on the limitations of our implementation of Equation 1.

### B.3 PROOF OF SUFFICIENT PRECISION OF APPROXIMATION

Referring to the content of the paragraphs just before and after Equation 17, the approximation in Equation 17 is more precise in the case of $C_i'(x) \cong \mathcal{AVG} \, C_i'(x) = C_i$ than in the case of approximating $C_i'(x) \cong C_i'(a)$. The average Jacobian of each term in Equation 1 can be denoted as $\mathcal{AVG} \, C_i'(x)$, while the Jacobian of each term in Equation 2 can be denoted as $C_i'(a)$. For the average Jacobian $\mathcal{AVG} \, C_i'(x)$, a better estimation in Equation 17 is obtained, as stated in the text near the equation. Consequently, applying Equation 17 to approximate Equation 13 results in a higher precision in estimating Equation 1 when averaging each Jacobian term separately, compared to utilizing the Jacobians without averaging. Therefore, a better approximation of the accurate average gradient is obtained compared to using the gradient. $\square$ The average gradient is proportional to the change in loss after the corresponding parameter update (Equation 14). Therefore, approximating the average gradient more precisely than current gradient-based methods can lead to more efficient minimization of batch loss, for example, by using Eq. 1. Therefore, learning can be enhanced compared to the potential of gradient-based methods.

## C  TEST LOSS CURVES OF MODEL B

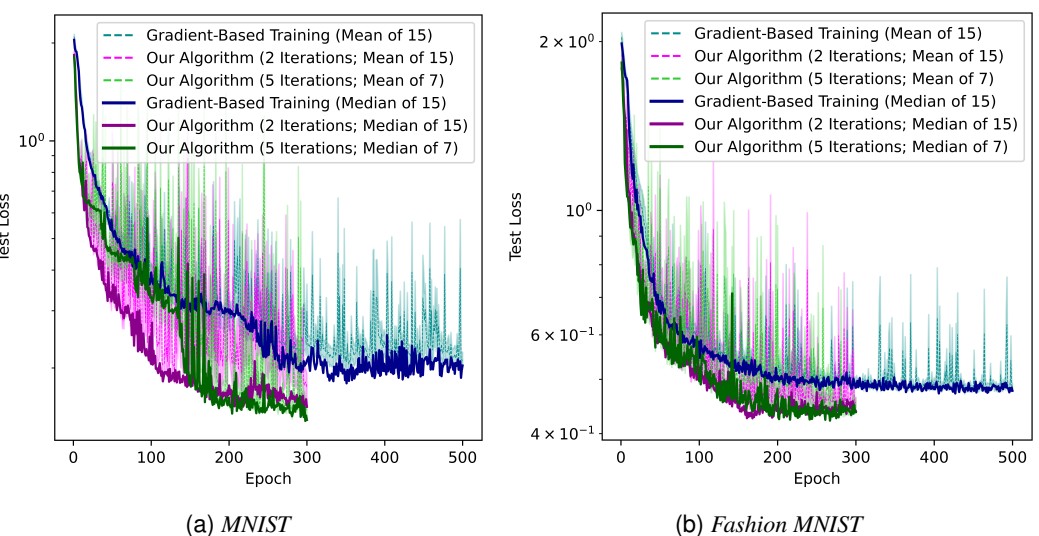

(a) *MNIST*
(b) *Fashion MNIST*

Figure 3: *Test losses of Model B*. Only mean curves contain confidence ranges (SEM).

## D  TEST ACCURACY CURVES OF MODEL B

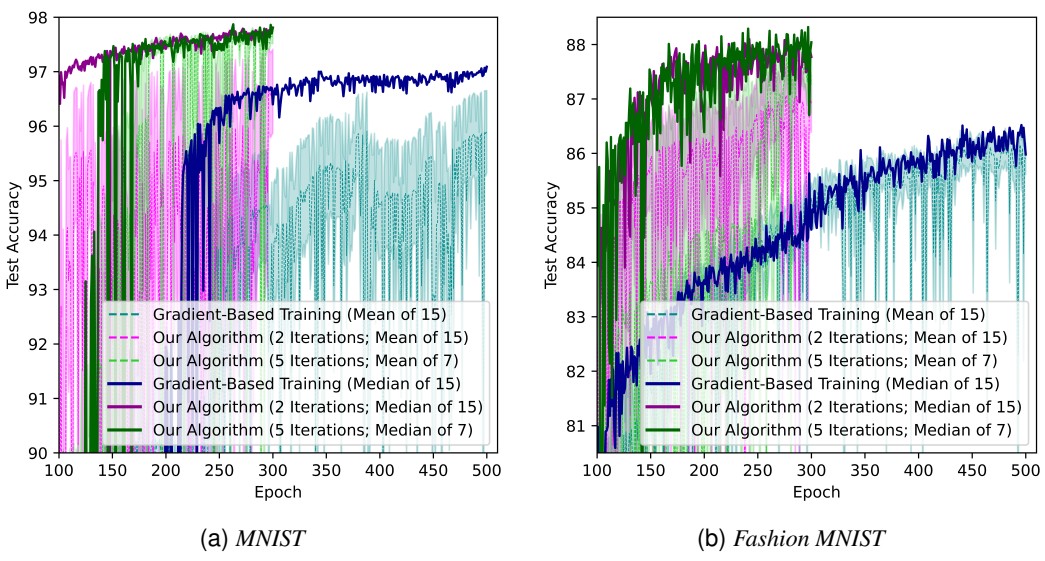

(a) *MNIST*
(b) *Fashion MNIST*

Figure 4: *Test accuracy of Model B*. Only mean curves contain confidence ranges (SEM).

## E  SCALING IN TERMS OF MODEL DEPTH

The algorithm based on the average gradient aims to reduce errors of the predicted influence on loss of a parameter update. In the case of the gradient-based approach, the errors arise from the impaired prediction of how inputs to subsequent layers influence their outputs (Fig. 1). Let us model the errors as multiplicative, because each time a fraction of output may be influenced by the error (Balduzzi et al., 2017). Therefore, when compared to the gradient-based algorithm as a baseline, the multiplicative errors are reduced after backpropagation through each nonlinear layer

(by computing the average Jacobian of the layer). Consequently, the incorporation of the average gradient exponentially reduces the error in terms of a count of nonlinear layers (that are involved in the backpropagation process). This explains the huge performance-improvement gap between the models for the method based on the average gradient, which emerges from the difference in models' depths. However, the gap is also increased due to the linearity of the ELU activation function in most of its domain, where the gradient equals its average. In this case, our algorithm produces results similar to those of gradient-based optimization.

If the errors (of the predicted influence on loss of a parameter update) are enormous, then the learning is impossible. Therefore, the learning performance tends to decrease after the error reaches a certain value for a given model, learning rate, and other parameters. From that point onward, our algorithm more efficiently reduces the batch loss compared to the gradient-based approach by minimizing the error in the loss-influence prediction. Importantly, the improvements tend to increase with both the number of nonlinear layers in a model and the learning rate.

## F    THREE-DIMENSIONAL COMPARISON OF THE GRADIENT AND THE AVERAGE GRADIENT

In our experiments, during a parameter update, in terms of the average reduction of loss for a batch, our algorithm lies between the gradient (red arrows in Figure 5) and the lowest average gradient (black arrows in Figure 5). Our algorithm does not always find a locally optimal solution (the best in the range of a single parameter update) because:

    a  The average gradient is approximated (by using Equation 1 instead of Equation 3, Equation 6 as a substitute of Equation 5, and the non-averaged gradient of the loss with respect to the last layer output).

    b  The optimal parameter update may be inaccurately estimated before the average gradient for this parameter update is calculated. Moreover, even after many iterations of Algorithm 4 (Appendix G), the update step may not converge to a locally optimal solution (black vectors in Figure 5).

    c  After the first iteration of our algorithm, only the negations of the directions of changes in each parameter are possible. Thus, the search for locally optimal updates is bounded by $2^{|\Theta|}$ combinations, where $|\Theta|$ is the count of trainable parameters.

Nevertheless, the $\mathcal{RD}$ metric (defined in Equation 7) indicates our algorithm minimizes the batch loss more efficiently on average compared to the gradient-based approach.

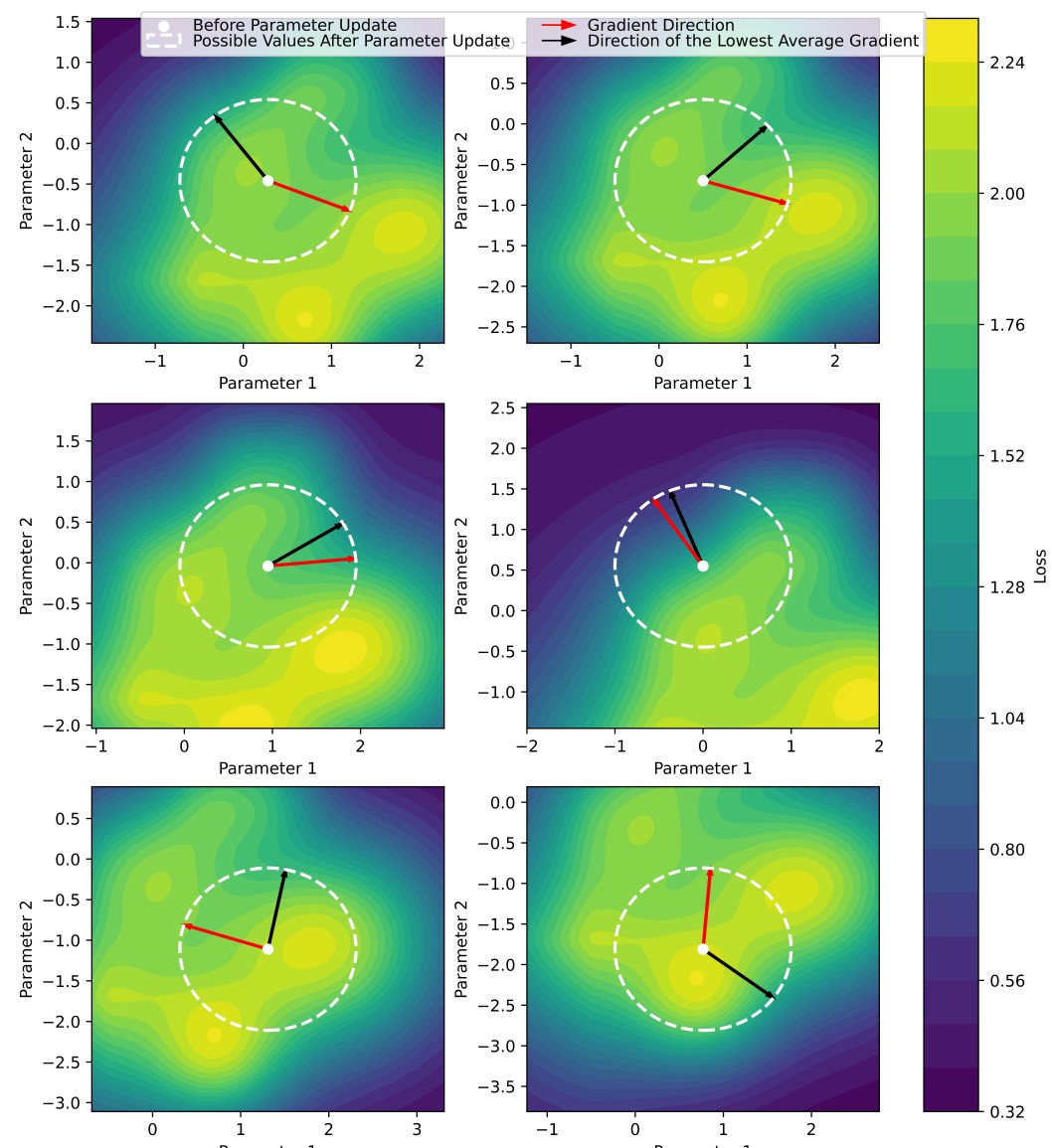

Figure 5: *Three-dimensional comparison of the gradient and the lowest average gradient in a few example scenarios.* The latter accurately reflects the influence on the loss of a parameter update. Furthermore, it accurately shows how each model parameter individually contributes to the change in the batch loss (Equation 14), which is utilized by our algorithm. Each plot illustrates the loss in terms of two example model parameters, assuming a specific magnitude for each parameter update (represented by the radius of each white circle). The arrows point to the loss values after an update based on the gradient and the average gradient. The average gradient is calculated for the update that minimizes it. Therefore, it points to the minimum loss on each white circle, although this minimum is not always achieved by the approximated average gradient computed by our algorithm.

# G ALGORITHM VERSION WITH PARAMETERIZED NUMBER OF ITERATIONS

---

**Algorithm 4** *Algorithm Version with Parameterized Number of Iterations* (two or more). The number of iterations is equal to the number of backpropagation calls and inferences in optimal implementation. The memory requirement of the ideal implementation would be higher than that of Adam by only an additional scalar size per parameter of the model.

---

**Input:** $model$: Neural Network Model to Train
    $dataset$: Training Dataset
    $lossFn$: Loss Function
    $optimizer$: Optimizer
    $iterCount$: Number of Backpropagation Iterations
**for all** $batch \in dataset$ **do**
    $modelInitial \leftarrow model$
    $modelCopy \leftarrow model$
    $initialOutput \leftarrow modelCopy(batch.\text{x})$ {It is assumed that $modelCopy$'s layers' results are kept inside $modelCopy$}
    $initialLoss \leftarrow LossFn(initialOutput, batch.\text{y})$
    $Backpropagate(initialLoss)$ {Compute the gradients using the standard backpropagation procedure. Assume that the gradients are stored inside $modelCopy$}
    $optimizer.\text{Step}(modelCopy)$ {Parameter update}
    $modelOutputAfterUpdate \leftarrow modelCopy(batch.\text{x})$
    $modelLossAfterUpdate \leftarrow LossFn(modelOutputAfterUpdate, batch.\text{y})$
    **for** $iter = 1, ..., iterCount - 1$ **do** {Loop ($iterCount - 1$) times, because one backward propagation is done}
        **if** $iter \neq 1$ **then**
            $modelCopy \leftarrow model$
            $modelCopy(batch.\text{x})$ {For each layer, compute its output, and store it inside $modelCopy$}
            $model \leftarrow modelInitial$
        **end if**
        $initialOutput \leftarrow model(batch.\text{x})$ {This computation is redundant if layer outputs are copied from $modelInitial$}
        $initialLoss \leftarrow LossFn(initialOutput, batch.\text{y})$ {Analogously, this computation is also redundant}
        **AveragedBackpropagation**$(model, modelCopy, initialLoss)$ {The procedure is described as Algorithm 2. The parameters of the $model$ are modified within}
    **end for**
**end for**

---

# H CONVOLUTIONAL NEURAL NETWORK ON IMDB

## H.1 METHODS

We refer to Model C (Tab. 4) as our very deep convolutional model, which we tested on the IMDB dataset. This model, primarily composed of convolutional layers, is designed to evaluate the performance of our learning algorithm on a deep convolutional neural network without skip connections. Skip connections simplify the learning task by enabling the network to leverage features that can be extracted by shallower networks (Veit et al., 2016). Our primary goal is to assess the algorithm's capabilities, rather than achieving state-of-the-art results.

IMDB preprocessing includes: (a) Equal split for test and training sets. (b) Duplicate removal. (c) Punctuation removal. (d) Tokenization. (e) Padding to the length of 122 (mean training-example length), and keeping the final part of each review. (f) Lemmantization. (g) Vectorization using GloVe embeddings (Pennington et al., 2014). Finally, the input has the shape $(1, 122, 50)$, where each input word is converted into its corresponding GloVe embedding with a length of 50. Model C (Tab. 4) utilizes multiple convolutional layers of shape $(1 \times 1)$, which are used to change the data shape and, for each "pixel", to extract features from the outputs of different filters. The neighboring dimensions of each GloVe embedding do not have any special relationship compared to the distinct

Table 4: *Model C.*

| Layers | Output Shape | Parameter Count |
|---|---|---|
| Convolution 2D $(1 \times 50)$, Tanh | $(50, 122, 1)$ | 2550 |
| Convolution 2D $(1 \times 1)$, Tanh | $(40, 122, 1)$ | 2040 |
| Convolution 2D $(1 \times 1)$, Tanh | $(35, 122, 1)$ | 1435 |
| Convolution 2D $(1 \times 1)$, Tanh | $(30, 122, 1)$ | 1080 |
| Convolution 2D $(1 \times 1)$, Tanh | $(27, 122, 1)$ | 837 |
| Convolution 2D $(1 \times 1)$, Tanh | $(24, 122, 1)$ | 672 |
| Convolution 2D $(1 \times 1)$, Tanh | $(21, 122, 1)$ | 525 |
| Convolution 2D $(1 \times 1)$, Tanh | $(18, 122, 1)$ | 396 |
| Convolution 2D $(1 \times 1)$, Tanh | $(16, 122, 1)$ | 304 |
| Convolution 2D $(1 \times 1)$, Tanh | $(14, 122, 1)$ | 238 |
| Convolution 2D $(1 \times 1)$, Tanh | $(12, 122, 1)$ | 180 |
| Convolution 2D $(1 \times 1)$, Tanh | $(10, 122, 1)$ | 130 |
| Convolution 2D $(1 \times 1)$, Tanh | $(8, 122, 1)$ | 88 |
| Convolution 2D $(1 \times 1)$, Tanh | $(6, 122, 1)$ | 54 |
| Convolution 2D $(1 \times 1)$, Tanh | $(5, 122, 1)$ | 35 |
| Convolution 2D $(3 \times 1)$ with stride $= 2$, Tanh | $(5, 60, 1)$ | 80 |
| **$25\times$Convolution 2D $(3 \times 1)$, Tanh** | $(5, 10, 1)$ | **$25\times 80$** |
| Flatten | 50 | |
| Linear, Tanh | 25 | 1275 |
| Linear, Tanh | 13 | 338 |
| Linear, Tanh | 7 | 98 |
| Linear, Tanh | 4 | 32 |
| Linear, Softmax | 2 | 10 |
| | | 14397 |

ones. Therefore, we used convolutions with a filter-size dimension equal to either one or all features in the GloVe embeddings.

## H.2 RESULTS

Both versions of our algorithm were more sample-efficient than the gradient-based RMSProp, as indicated by the training loss (Fig. 6). In the case of gradient-based training, the trade-off between the mean and median of the training loss is visible in both Figs. 6a and 6b. The tendency for instability in training with a higher learning rate leads to the occurrence of outliers, also in terms of whole worse trainings, which lower the mean. However, the median is resistant to these outliers. This can also be observed in Figures 7b and 8b. To evaluate both the mean and median, considering the trade-off between them, we compared the methods by averaging the median and mean losses. This approach provides a consistent comparison result across both learning rates of the gradient-based RMSProp (Fig. 6). Using this evaluation method, the performance of gradient-based RMSProp at epoch 200 is approximately equal to the results of the two iterations of our method at epochs 125 and 130, in Figs. 6a and 6b, respectively. This translates to a sample efficiency between 53% and 60% higher in favor of the two iterations of our algorithm compared to the vanilla RMSProp. Surprisingly, the performance of the four iterations falls between the other methods, with the sample efficiency gain ranging from 25% to 30%. The $\mathcal{RD} = 0.0394 \pm 0.0053$ metric also favors the two-iteration variant, outperforming the $\mathcal{RD} = 0.0210 \pm 0.0018$ achieved by the four iterations.

The test-accuracy (Fig. 8) and test-loss curves (Fig. 7) should be interpreted in the context that the objective of the hyperparameter search is dependent solely on the training loss. In addition, considering the trade-off between the mean and median losses, which occurs between the lower and higher learning rates of the vanilla RMSProp, slightly better mean generalization in the gradient training for the low learning rate (Fig. 7a) does not imply generally better test performance. However, the comparison using the same learning rate (Fig. 7b) indicates that the two-iteration variant achieves the most stable test-loss performance.

Table 5: *Learning rates* for the IMDB dataset. The "Learning Rate" column presents the final chosen learning rates for the experiments. A repeated loss of $0.6931$ is equivalent to the lack of training. The table includes the results of the final experiments; however, the results are clipped to 150 epochs for the variants of our algorithm. The best results and the learning rates chosen for the experiments are marked in bold.

| Method | Learning Rate | Most Important Hyperparameter Search Results [Learning Rate: Avg. of Min. Training Loss] |
|---|---|---|
| **RMSProp** (200 epochs) | **3.641e−4** (stable trainings, small number of outliers, low average losses) and **6.906e−4** (slightly higher average training losses, but lower median losses) | $9.545e{-}5 : 0.6719$; $1.193e{-}4 : 0.5605$ 
 $1.491e{-}4 : 0.4121$; $1.864e{-}4 : 0.5106$ 
 $2.330e{-}4 : 0.3567$; $2.912e{-}4 : 0.4026$ 
 **3.641e−4 : 0.3370**; $4.551e{-}4 : 0.3407$ 
 $5.689e{-}4 : 0.5243$; **6.906e−4 : 0.2801** 
 $8.384e{-}4 : 0.4414$; $1.018e{-}4 : 0.4565$ 
 $1.236e{-}3 : 0.6931$; $1.500e{-}3 : 0.6931$ 
 Repeated trainings: 
 **3.641e−4 : (0.4307 ± 0.0143)**; (30 trainings) 
 **6.906e−4 : (0.4435 ± 0.0154)**; (50 trainings) |
| **2 Iterations** (150 epochs) | **6.906e−4** (low training losses, easy to compare with **RMSProp** due to matching learning rate) | $3.641e{-}4 : 0.4948$; **4.733e−4 : 0.3373** 
 $6.153e{-}4 : 0.4385$; $7.999e{-}4 : 0.4316$ 
 $1.040e{-}3 : 0.4096$; $1.352e{-}3 : 0.6931$ 
 Repeated trainings: 
 **4.733e−4 : (0.4574 ± 0.0127)**; (15 trainings) 
 **6.906e−4 : (0.4225 ± 0.0125)**; (30 trainings) |
| **5 Iterations** (150 epochs) | **6.906e−4** (easy to compare with **RMSProp** and **2 Iterations** due to matching learning rates) | **4.734e−4 : 0.3766**; $6.153e{-}4 : 0.4410$ 
 Repeated trainings: 
 **6.906e−4 : (0.4567 ± 0.0129)**; (15 trainings) |

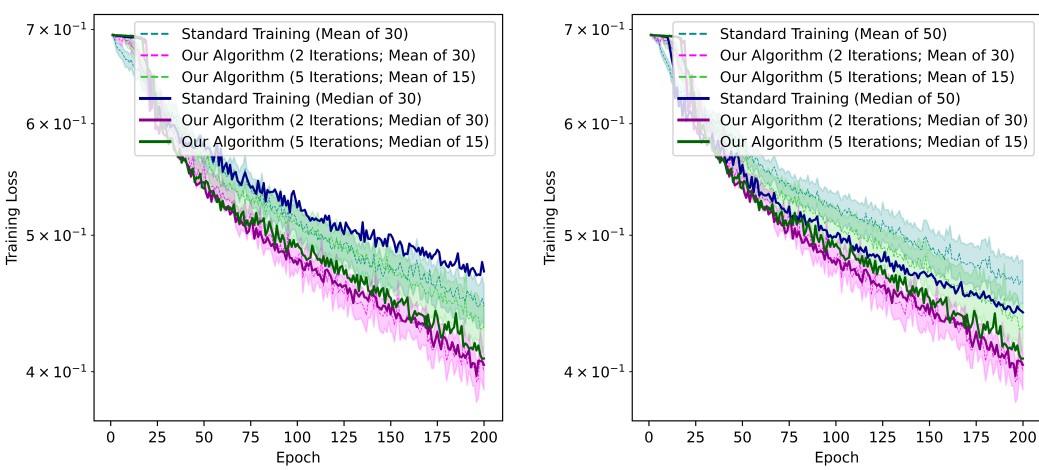

(a) Gradient-based RMSProp with the lower learning rate of **3.641e−4**.

(b) Gradient-based RMSProp with the lower learning rate of **6.906e−4**.

Figure 6: ***Training loss*** *of Model C*. Only mean curves contain confidence ranges (SEM).

Due to suboptimal backpropagation of the average gradient through activations in our implementation, it has a bigger computational overhead for models applying activations to large feature maps. Therefore, our implementation is computationally slower relatively to the gradient-based training for Model C than in the case of Model B.

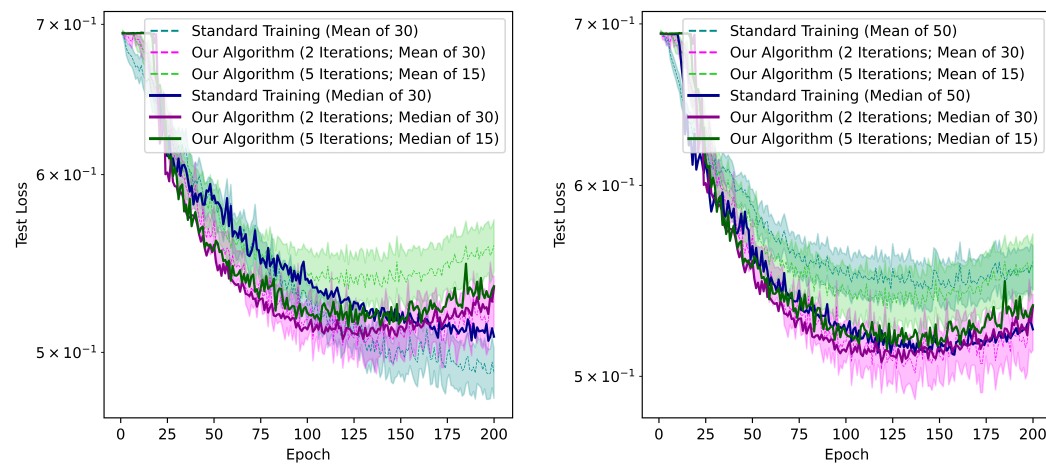

(a) Gradient-based RMSProp with the lower learning rate of **3.641e−4**.

(b) Gradient-based RMSProp with the lower learning rate of **6.906e−4**.

Figure 7: ***Test loss*** *of Model C*. Only mean curves contain confidence ranges (SEM).

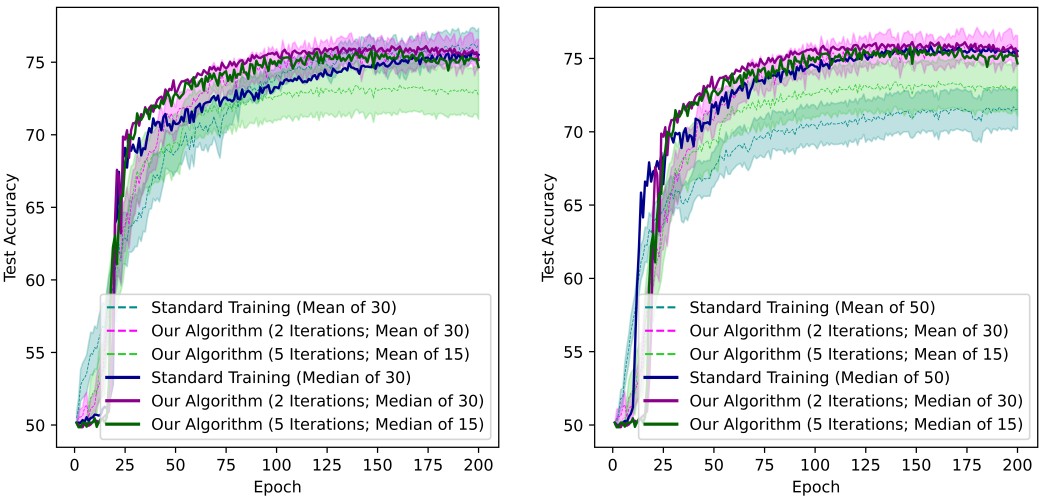

(a) Gradient-based RMSProp with the lower learning rate of **3.641e−4**.

(b) Gradient-based RMSProp with the lower learning rate of **6.906e−4**.

Figure 8: ***Test accuracy*** *of Model C*. Only mean curves contain confidence ranges (SEM).

## I  EXPERIMENTS WITH ALTERNATIVE WEIGHT INITIALIZATION FOR MODEL B

We repeated the experiments for Model B on the MNIST and Fashion MNIST datasets due to suboptimal parameter initialization, which resulted in vanishing gradients at the start of the training. During the repeated experiments, we initialized the weights using the Glorot uniform method (Glorot & Bengio, 2010), which is specifically designed to initialize layers with nonlinear activations such as Tanh or Sigmoid. A gradient-magnitude gain of $\frac{5}{3}$ for Tanh activations was used, as recommended by the PyTorch library. Biases were initialized to zero. The gradient magnitudes were examined to ensure they fell within a satisfactory range after initialization. Training length was reduced to 125 epochs for the gradient-based RMSProp and 50 epochs for two iterations of our method to test a 2.5x learning speedup.

As expected, similar magnitudes of learning rates performed well in the trainings using different weight initializations (compare Tables 3 and 6). The losses using our method are significantly lower

Table 6: *Results for different learning rates.*

| Dataset | Method | The Most Important Results [Learning Rate: Avg. of Min. Training Loss] |
|---------|--------|------------------------------------------------------------------------|
| MNIST | **RMSProp** (**125** epochs) | 8 trainings per each learning rate: 
 $1e{-}4 : 0.0945 \pm 0.0062$;  $1.5e{-}4 : 0.0821 \pm 0.0121$; 
 $1.75e{-}4 : 0.0750 \pm 0.0102$;  $2e{-}4 : 0.0519 \pm 0.0057$; 
 $2.25e{-}4 : 0.0862 \pm 0.0171$;  $\mathbf{2.5e{-}4 : 0.0417 \pm 0.0035}$; 
 $2.75e{-}4 : 0.0515 \pm 0.0094$;  $3e{-}4 : 0.0749 \pm 0.0187$; 
 $3.5e{-}4 : 0.0536 \pm 0.0095$;  $4e{-}4 : 0.0604 \pm 0.0188$; 
 50 trainings: $\mathbf{2.5e{-}4 : 0.0478 \pm 0.0032}$; |
| | **2 Iterations** (**50** epochs) | 3 trainings per each learning rate: 
 $1.54e{-}4 : 0.128 \pm 0.0068$;  $4.61e{-}4 : 0.0351 \pm 0.0029$; 
 8 trainings per each learning rate: 
 $6e{-}4 : 0.0320 \pm 0.0013$;  $8e{-}4 : 0.0256 \pm 0.0019$; 
 $\mathbf{9e{-}4 : 0.0245 \pm 0.0005}$;  $1e{-}3 : 0.0255 \pm 0.0021$; 
 $1.1e{-}3 : 0.0253 \pm 0.0016$;  $1.2e{-}3 : 0.0246 \pm 0.0005$; 
 $1.4e{-}3 : 0.0255 \pm 0.0002$; |
| Fashion MNIST | **RMSProp** (**125** epochs) | 8 trainings per each learning rate: 
 $2.5e{-}4 : 0.329 \pm 0.022$;  $3e{-}4 : 0.352 \pm 0.024$; 
 $\mathbf{3.5e{-}4 : 0.315 \pm 0.025}$;  $4e{-}4 : 0.347 \pm 0.024$; 
 $4.5e{-}4 : 0.374 \pm 0.029$;  $5e{-}4 : 0.396 \pm 0.025$; 
 $\mathbf{5.5e{-}4 : 0.310 \pm 0.024}$;  $6e{-}4 : 0.371 \pm 0.025$; 
 $6.5e{-}4 : 0.368 \pm 0.034$;  $7e{-}4 : 0.407 \pm 0.021$; 
 50 trainings: $\mathbf{3.5e{-}4 : 0.344 \pm 0.010}$; |
| | **2 Iterations** (**50** epochs) | 3 trainings per each learning rate: 
 $1.33e{-}4 : 0.467 \pm 0.002$;  $1.75e{-}4 : 0.463 \pm 0.001$; 
 8 trainings per each learning rate: 
 $6e{-}4 : 0.269 \pm 0.003$;  $8e{-}4 : 0.255 \pm 0.002$; 
 $9e{-}4 : 0.255 \pm 0.002$;  $1e{-}3 : 0.254 \pm 0.003$; 
 $1.1e{-}3 : 0.253 \pm 0.003$;  $\mathbf{1.2e{-}3 : 0.245 \pm 0.001}$; 
 $1.4e{-}3 : 0.252 \pm 0.003$;  $1.5e{-}3 : 0.261 \pm 0.004$; 
 $1.6e{-}3 : 0.253 \pm 0.003$;  $1.8e{-}3 : 0.273 \pm 0.008$; 
 $2e{-}3 : 0.262 \pm 0.000$; |

after 2.5 times fewer epochs. Therefore, the two iterations of our method increase sample efficiency by more than 2.5 times. Good performance of the method across different parameter-initialization distributions is essential for its practical application as it contributes to robustness. Importantly, our algorithm maintains its performance gain compared to the gradient-based RMSProp in scenarios involving vanishing gradients, as demonstrated in the main experiments.

## J   FUTURE WORK

Interesting directions for further experiments include: (a) Computing the average gradients over a much larger range than that of a parameter update to capture the global trend of the loss landscape. (b) More accurate approximation of the average Jacobians using Equation 5 instead of Equation 6. This would enable computing the average Jacobians of linear operators. Therefore, the algorithm based on the average gradient may enhance trainings of deep models without nonlinear activations. Moreover, the usage of Equation 5 may further improve the performance in the case of many nonlinear activations because of the increased precision in approximating the average gradient. (c) Incorporation of the momentum into our algorithm. Preferably Nesterov momentum (Dozat, 2016) should be used. If not, the average gradient would also be calculated for the momentum part of the update step. This could often reverse the direction of the momentum for a model parameter, thereby impairing the effectiveness of the entire momentum procedure.   (d) Development of similar algorithms, but with update steps, that, for a given model parameter, vary in size over the iterations

of the average-gradient computation. By adjusting the step size of each model parameter to the absolute value of the average gradient, the learning process may be enhanced. (e) Tests of the method on large and very deep architectures, that are used in practice and contain many nonlinear layers. (f) More research on how the method scales up (Appendix E), also in relation to the number of neurons in layers of neural networks. (g) Experiments with learning without forgetting (Li & Hoiem, 2017) and online learning. Sample efficiency may be very beneficial there.

