# OpenReview forum: "Efficient Gradient-Based Algorithm for Training Deep Learning Models With Many Nonlinear Activations"
_ICLR.cc/2025/Conference — Submitted to ICLR 2025_

### Official Review · Reviewer_bUVm · 2024-10-29

**Soundness:** 2
**Presentation:** 3
**Contribution:** 2
**Rating:** 5
**Confidence:** 2

**Summary:**

The authors describe a modification to how gradients are computed for first order optimizers that uses a gradient-averaging technique that they claim generalizes better, would require less time in optimal implementation, is more robust over choice of learning rate, and improves sample efficiency according to their empirical benchmarks. They prove that the average gradient is directly proportional to the loss delta in their appendix.

S1.2 describes prior works and trends, delineating the work from momentum-based methods which maintain a moving average of gradients and from averaging gradients across a batch, which is common place. The authors instead are averaging the gradient over the range between the current parameter vector ($\theta_t$) and the parameter vector given by a vanilla RMSProp gradient updates step ($\theta_{t+1}$ or more commonly $\theta$' in this paper); as the authors point out, this average gradient can be computed by evaluating the model at $\theta_{t}$ and $\theta_{t+1}$ and dividing by the range.

They describe in great detail the average gradient and provide one-dimensional intuition (which is very appreciated). The algorithms are described in similar detail. Equation 7 describes a novel metric that the authors will use to evaluate their experiments, called the RD or relative difference, which measures the difference in change of loss at each step of the average gradient vs the vanilla gradient, normalized to the change in loss of the vanilla gradient. In Table 2, the authors argue that the average gradient is more robust over learning rates than vanilla gradients.

They proceed to empirical methods and show that while the average gradient essentially does not improve upon vanilla gradients for a LeNet style architecture, it does seem to show improvements on a small sample of experiments using an (as the authors admit) impractical network architecture of 30 layers (mostly composed of repeated Linear layers of 10x10 size) on the MNIST and FMNIST dataset. The RD metric is used to argue that the average gradients of the architecture B significantly outperforms vanilla gradients. The visualize the loss cuvres of these experiments which corroborates that Model B does appear to train faster in terms of sample efficiency.

The authors conclude by restating their main claims.

**Strengths:**

I think the presentation of the work is mostly good; I applaud the authors for attempting an innovative solution and attempting to provide both a theoretical and empirical foundation for their work. Clearly if the average gradient can be definitively shown to robustly outperform the vanilla gradient the work would be of tremendously high impact and appropriate for a major ML conference. I think if the authors were to submit a major revision which addresses my worries specifically regarding the empirical experiments, the paper could be accepted. As is, I think it is clearly a not accept.

I appreciate the authors making their code available and endeavoring to put their algorithms to great detail into their work. Reproducibility is important.

**Weaknesses:**

I believe that the RD metric is computed per step for each run, which leads to a large number of samples and is where you draw your ability to separate the performance between average and vanilla gradients. If that is correct, the RD metric samples being collected are not independent of one another in the context of a single experiment generating sequential gradient steps. Because of this, I do not think the empirical argument of demonstrated performance gain is sound. There are multiple ways to address this: I think one way would be to run the experiments many times with different seeds for initialization + batch order; then for each experiment, fit a power law curve (or similar) to the observed loss, collect the power law coefficients and perform your statistics on those samples.

You write multiple times that the average gradient provides more information on the change in loss caused by the associated parameter update of a model; this is in fact the first claim in your abstract. From this I think you mean that you have proven that the average gradient can be shown to be proportional to the loss differences; however it seems like a logical leap to me that simply because the average gradient is proportional to loss deltas that this implies it is a robust improvement over vanilla gradients.

One cynical perspective is that the vanilla RMSProp performed as well as your averaging gradient on model A which is similar to common CNN architectures, but only on the model that is as you admit impractical (26 x linear layers of shape 10 x 10) does your average gradient outperform. I am confused as to why you would include experiments on an impractical network and furthermore only include as a baseline the performance of RMSProp on that same impractical network. Overall I do not find the empirical evidence presented very conclusive that the average gradient in the context of models of great interest to the community.

Minor: I would change the color of the point representing "before parameter update" figure 1 as the grey does not appear well in my print out or when viewed online. Perhaps consider a black star? Also on figure 1, what is the difference between the left and right plot represent? They should be titled.

the authors point out that there is an increased cost to running the average gradient that grows with the number of iterations, growing from 3x for 2-iterations and 8x for 5x iterations. I wanted to ask if this is a fundamental limitation or merely a limitation of the current implementation? If this can not be conceivably improved, it begs the question of whether vanilla gradients might be less sample efficient but overall more time (and therefore cost!) efficient. I think you are arguing through something you call the "optimal implementation" that you are in fact faster. Can you re-explain what this optimal implementation is? Is it a learning rate choice, is it a n-iterations choice?

**Questions:**

In figure 1, left side, you show how the gradient line is negative sloped and the average gradient line is positive sloped. The local minima of the loss function visualized is in the positive direction relative to the "before parameter update" point. A negative gradient is needed to move the parameter update using the traditional GD update rule: ($\theta_{t+1} = \theta_t - \eta \frac{\partial l}{\theta_t}$) in the positive direction, but seemingly replacing the gradient with the average gradient would give a positive slope moving the point in the wrong direction. Can you explain what the reader is meant to take away from figure 1? What is wrong with my thinking about this?

Related to the above, you include an appendix titled "Loss minimization potential" but I believe the main proof is that the average gradient is proportional to the loss delta. I don't think that necessarily shows that the average gradient step would lead to loss minimization? More precisely, in figure one you seem to show a step where the average gradient is in the wrong direction, and if in fact if the loss delta is zero, then you seem to prove that the average gradient is zero and no update occurs -- i.e., if the chord intersecting the loss function is zero sloped, no update occurs. Where am I going wrong? Is it that the "n-iterations" are the number of gradients being averaged and so in practice these chords are unlikely to arise?

I guess related to the above, what specifically are the "n-iterations"?

In your introduction you reason that it is most fair to compare your research to models trained with RMSProp since by default the momentum of the RMSProp optimizer is zero and therefore comparing losses after each batch update is valid. Can you explain this reasoning and why comparing to SGD with momentum chosen to be zero, or Adam with momentum chosen to be zero would be less valid? Or more generally why comparing to optimizers that use momentum is invalid?

line 141; "Then it is assumed that the absolute value of the parameter delta of the RMSProp optimizer is good enough to retain it" can you explain what the assumption is and whether it is specific to RMSProp or needs to be assumed of other optimizers as well?

In historical works comparing optimizers, was the relative difference or (RD, equation 7) metric that you propose used? Are you the first to propose it? Why did you choose to use this metric, and are there other important measures that might matter to users? For example, RD treats steps that might take different amounts of compute time but cover the same number of update steps equally. How much longer do individual steps take with average gradient compared to vanilla RMSProp? How does this scale with the number of parameters, for example, can you roughly guess what hardware (specifically how much memory) and how much time would be required to run your algorithm on a model of 30 layers like model B but with order 1 Million model parameters?

---

> ### Author Response · Authors · 2024-11-17
> **Answers to the first 4 questions**
>
> Answers to the questions (your paragraphs and questions are numbered in the format [paragraph number].[question number]):
>
> 1.1: The main idea conveyed by Fig. 1 is to show that the average gradient is proportional to the loss delta (caused by a single parameter update, not the whole update), whereas the gradient is not.
>
> 1.2 (and the whole paragraph): You probably assume that the aim of the updates in the figure is to reach the local minimum, whereas it is the opposite. That particular plot shows that the gradient update causes the increase in the loss, while the average gradient beforehand provides the information, that the loss is increased (so the update of that parameter can be, for example, zeroed or negated). This is particularly important when some $n>1$ parameters are updated, because then, using only the gradient, there is no way to determine whether a given update of a single parameter contributed to the loss increase or decrease (when the learning rate is big, and we want it to be big to converge faster). The **huge** advantage over the gradient.
>
> Would you like to have it clearer on the plot, that the minimum is local not global?
>
> The text before 2.1: From the point of the contribution, by far more important is the B3 subsection rather than the loss proportionality in B1. B3 means that the average gradient can be efficiently computed, while B1 is more obvious.
> 2.1: As it is stated in the paper, the algorithm is iterative. Each iteration optionally negates the direction of each single-parameter update. That iterative procedure has at least a very good convergence, which is shown in the experiments. Moreover, it can be intuitively explained, that when one update direction of a parameter contributes to the increase in a loss (which is the first iteration of our algorithm), then checking the other update direction (which is the second iteration) may fix the loss, which is more or less obvious that it **statistically** reduces the loss more than the first iteration (the regular gradient update).
>
> 2.2: Partly answered in 2.1 and 1.2. The zero-sloped chord intersecting the loss function is a very specific example, where the gradient update (SGD w/o momentum or RMSProp) is the same as for our algorithm and equals zero. If it didn’t behave in that way, it could for example tend to get out of the global minimum point once reaching it.
>
> 2.3+3.1: Each of the $n$ iterations include computing the average gradient (but for the first iteration only the gradient), and then during every iteration the update directions for parameters are optionally negated to statistically reduce the batch loss (answer 2.1), but the average gradient changes after negating that update directions, so virtually the new ‘chord’ (one per parameter, represented by the average gradient) is computed for the optionally negated update directions to again refine the update direction every iteration. One iteration (except for the first one) basically consists of computing a ‘chord’ for each parameter (the average gradient) and then eventual negations of update directions  are performed for some parameters to reduce the estimated average gradient (which turns into loss reduction because the average gradient is proportional to loss change after update according to Eq. 14. Appendix B3 contains the proof that the estimation of the average gradient in Eq. 1 is sufficient).
>
> 4.1: **Adam without momentum is RMSProp**, while SGD without momentum is an option for experiments, but it is relatively rarely used + we wanted to test how the algorithm behaves on adaptive update steps (like RMSProp or Adam). However, in the specific line that we write about that the comparison would be less valid, we write it in the context of RD metric. It measures loss reduction on a batch per a single update, so it is suitable for optimizers that also reduce the loss of a batch per a single update. We state the context in the first part of the sentence.
>
> 4.2: To prevent the comparison of momentum to no momentum to be mixed with the comparison of gradient RMSProp to average gradient RMSProp because the momentum may give additional advantage for momentum optimizers, and we would rather implement momentum to our method and then compare. It would be more informative. There is a way of implemeting the momentum into our method that is not mathematically obvious and may give better results than the vanilla momentum (similar trick to the one with average gradient).

---

> ### Author Response · Authors · 2024-11-17
> **Answers to the questions 5-6**
>
> 5.1: Intuitive argument: The assumption is because the magnitude of the RMSProp update reflects parameter-update importance, so we can keep it invariant per update and take it from the RMSProp, which is somehow good at estimating the importance. Keeping the two possibilities for update of each parameter (+ or - |RMSProp update|): a) prevents value explosions and provides stability, b) makes it easier to synchronize the average gradient (denoted further as AG) with parameter update over iterations (‘synchronization’ defined as: the final update must point to AG direction computed on the range of the final update). The assumption is invariant across the first-order optimizers without momentum.
>
> 6.1-3: RD is used because the math foundations of the algorithm (appendix B) prove efficient loss minimization for a **batch** (answer 4.2). RD is nothing more than comparing **batch** loss decreases. Therefore we used RD to validate the maths. BTW to summarize most of the important math of the paper: Actually, **appendix B proves that if we use Eq. 1 and synchronize AG with the update in a manner that AG points towards loss minimization for every model parameter, then RD is better than for the gradient.**
>
> 6.4-5: Speed approximations are mentioned in the paper. **Execution operations that take nearly all execution time**: assuming Vanilla RMSProp with $n$ params (and O(n) neurons), (execution order for a single batch):  1 forward O(n) +  1 backprop O(n) +  1 RMSprop step O(n). Average-Gradient RMSprop 2 iter, execution order:  1 fast params copy O(n) +  1 forward O(n) +  1 backprop O(n)+  1 RMSProp step O(n) + 1 forward O(n) +  1 only slightly slower averaged backprop O(n) +  1 Parameter update using the average gradient (faster than RMSProp, because RMSprop adaptive learning rate isn’t recomputed). So the average-gradient RMSprop with 2 iter is about from 2 to 2.333 times slower per step optimally (some of its subprocedures are faster than the other), which scales exactly the same for more parameters (complexity O(n)).  So even for 1*10^12 params it is up to 2.333 times slower. The accurate value depends on some hardware details, i.e. the ratio of GPU memory read time to the time of numerical operation on GPU/TPU registers.
>
> **Memory requirement** for vanilla RMSProp: ~2*n (+ dataset size or part of it), for AG RMSProp 2 iter: ~3*n (+ dataset size or part of it) (because of the model copy of n parameters).  AG RMSProp x>=3 iter: ~4*n  (+ dataset size or part of it).
>
> Despite the answers to the questions are more or less directly answered in the paper (sometimes in appendices or different sections than where the questions arise), please feel free to ask any further questions, or clarifications.

---

> ### Author Response · Authors · 2024-11-17
> **Answers to weaknesses and strengths**
>
> Our comments on strengths:
>
> We will try to provide an additional benchmark and a model before the deadline, that will be more ‘empirical’. However, our implementation does not allow for RNN yet, so it will be a very deep CNN.
>
> Our comments on the weaknesses (numbered by paragraphs):
>
> 1: **You are right that the RD metric is imperfect indicator of algorithm performance on the whole dataset** not a batch. Of course we do not state otherwise in the paper. The reason for the incorporation of RD metric is explained in our answer to your question no. 6.1-3. RD is a ‘sanity check’. We should clarify that in the paper. To compare the performance on the whole dataset (not a batch), we use min/max/avg/median training/test accuracy and loss.
>
> 2: **Yes, the proportionality to the loss is proven in Eq. 14.** So it is a strength rather than a weakness. Indeed, as you wrote, it is a robust improvement over vanilla gradients. Why is it in weaknesses?
>
> 3: This is initial phrase of the study, so we used simple models. Smaller improvements on model A are expected (Appendix E).
>
> 3 further: We would argue that the community is interested in the fact, that **a modification of gradient on nonlinear activations of very deep models may increase sample efficiency multiple times**, which is the direct conclusion of our experiments.
>
> We think that the community could be interested in the results on model B because it shows, that not all quite simple models with common activation functions (that are the most similar to biological activations working in ‘binary fashion’) can be trained with vanilla gradient, which is proven in our experiments to be far from the optimal algorithm to minimize the loss. Human brain has many thousands of neural layers, while the RMSprop algorithm fails to efficiently solve 30 layers of dense! This indicates, that the biological brain works in a distinct manner than pure gradient descent. We **prove and implement** a possible solution, that works for sure in some cases at least. Importantly, the empirically proven huge inoptimality of the vanilla gradient means, that historical ML-architecture searches were biased towards models with good behavior when trained with inoptimal training algorithm. Example: a lot of very deep models are trained much more efficiently and achieve better scores with residual connections, while if gradient descent was optimal to minimize the loss, many of them could **be trained** to behave in such manner, that the intermediate-layer data is passed through further layers, without skip connections. Therefore, the residual connections make huge difference to the vanilla gradient. Therefore historical ML-architecture searches based on gradient optimizers were biased towards residual connections and shallow models. Our experiment shows that, at least in some cases, the inoptimality of learning can be solved efficiently.
>
> It is also worth mentioning that if the sample efficiency according to median training loss increased three times for Model B, and even if the gain was 50 times smaller for some different very deep models (which is unlikely to happen for the same type of layers and activations), it would be still **significant achievement** for our algorithm. Also similar optimizers work well across domains, so a domain switch isn’t expected to bring significant degradation of the results. However, \text{we will try to finish hyperparameter search and tests on one more benchmark and a more practical model architecture until the deadline.**
>
> Moreover, we present a much, much faster algorithm than [1] and prove the performance guarantees of its core equation (Eq. 1), therefore our contribution is much bigger than [1].
>
> [1] Sundararajan, M., Taly, A., & Yan, Q. (2017, July). Axiomatic attribution for deep networks. In International Conference on Machine Learning (pp. 3319-3328). PMLR.
>
>
> 4: Thanks, we will clarify the illustration.
>
> 5.1: For the 2 iterations we answered in the response to your question no. 6.4-5. The fundamental runtime limitation of n-iteration for $n\ge 3$ is about $n\cdot R+n\cdot C$, where R is the time of vanilla RMSprop batch update, while C is the time needed to copy model parameters. For $n=2$ the limitation is about $2R+C$, but as mentioned in 6.4-5, some $R$ subprocedures can be slightly sped up, but also averaged backpropagation may be **slightly** slower. But $2R+C$ is a quite good approximation.
>
> 5.2-3: We refer to the optimal implementation as the one that optimally manages intermediate results (instead of recomputing them), don’t compute any costly statistics (like the loss after the final update) and is also optimized on the level of machine code to the same extent as the RMSProp from the libraries like PyTorch or Tensorflow. **The ‘optimal implementation’ doesn’t refer to a learning rate choice or the number of iterations choice.** It is logically the same algorithm returning the same results, but as fast implementation as it is possible.

---

> ### Author Response · Authors · 2024-11-17
> **Brief comment on summary**
>
> Answer to the statement: „as the authors point out, this average gradient can be computed by evaluating the model at $\theta_t$ and $\theta_{t+1}$ and dividing by the range”: To be strict, this is true only for a model with one parameter, or a function: $\mathbb{R}\rightarrow\mathbb{R}$. Otherwise, it would be only an arithmetic average of two losses (scalar). The average gradient is the loss-change contribution of **each** model parameter on a range (for example, of a weight update; Eq. 14). Therefore, the procedure is much more complicated (Pseudocodes 1-4).

---

> ### Comment · Reviewer_bUVm · 2024-11-19
> **Reply to some of these replies.**
>
> I plan to come back and address more aspects.
>
> Re. Figure 1: After your clarification I understand it now. Yes, I indeed assumed that the minima plotted was global.
>
> Re. your questions back about how to improve figure 1: Yes, I would suggest expanding out the loss function to show some other minima, then plot the "after average gradient update" point with the "after vanilla gradient update" point so the reader will clearly see how the average gradient is an overall improvement intuitively. Re. your point that the average gradient can be at times zeroed or negated, I would also specifically include suggestive examples where you would want to zero or negate the average gradient to incorporate this point... I suppose its possible the average gradient is zero'd in both of these examples, explaining why there is not a update?
>
> Re. the RD metric: I understand that the RD metric is computed per batch; my point is that you are computing it successively for each experiment (I suspect!). Re. this concern, can you expand on what you mean with: "Importantly, the same model parameters are set before weight updates of both methods." I'm interpreting this to mean, based off figure 2.c and 2.d, that the weights are synchronized for only the first update but no others? It could also mean, after every single weight update you re-synchronize the models?
>
> So assuming I'm right: Upon reflecting on this more, there are really two issues. After the first gradient update you are effectively comparing the loss deltas between two different parameters states. I think you want to show that for any common parameter state the loss delta from applying an average gradient update is larger than the loss delta of a vanilla gradient update. But since the two models are in different states after the first update, the RD metric is comparing apples and oranges-- it could be that the vanilla gradient applied to the model in the same state as the average gradient's current state would have an improved loss delta (empirically), but you did not test for it.
>
> Even if I'm not right above about your methods, this second issue would persist. The second issue is that you point out you are using an adaptive step size optimizer RMSprop. This mean that the learning rate is dependent upon previous updates, which means that there is not independence between each parameter update in a single model-training/experiment. If the parameter updates in RMSprop are not independent, then surely the loss deltas are not either, so neither are the RD metrics computed. If your sample of RD metrics are not independent then you violate an assumption when performing your statistic (which if I recall is just some central tendency + error?)

---

> ### Author Response · Authors · 2024-12-04
> **Reply**
>
> Hello, nice to read your reply! Thanks for your suggestions!
>
> Re. Fig. 1. Good point, we changed ‘After Parameter Update’ to ‘After Parameter Update Suggested by the Gradient’, now it is much more clear :)
>
> Re. Fig. 1. Before you commented we had already changed Fig. 1 (so we haven’t done it according to your last comment, but we used information from your previous comments). Now the marks are clearly visible, each subplot contains a short description which makes it easier to understand, local minimums are marked on the legends, and now it is clear that both of the subplots should be interpreted separately (and we changed ‘After Parameter Update’ on the legend to ‘After Parameter Update Suggested by the Gradient’).
> For the first subplot, the "after average gradient update" point depends on the number of iterations of our algorithm and the loss landscape on the other side of the gradient update. Of course, we can assume a particular loss landscape, but figures, just updated by us and hopefully clearer, contain all of the possibilities: 1) directions of the gradient and the average gradient are the same, 2) they are different. Therefore, they are quite general, don’t focus on a specific case, are kept very simple, stimulate imagination on how the average gradient can be utilized, and keep the abstraction separating our algorithm from the average gradient (that can be utilized in multiple ways).
> However, if you don’t like the current updated Fig. 1, let us know, we will change it :)
>
> Re. RD ISSUE 1. The ‘weight synchronization’ doesn’t affect Figs 2.c and 2.d in any kind (otherwise of course they may not accurately predict the performance gains). The first iteration of our method is the gradient-based RMSProp procedure, so the sentence „… the same model parameters are used before weight updates in both methods” means two methods: 1) our method and 2) the first iteration of our method which is the standard RMSProp procedure. Therefore we calculate the loss decreases without any manipulations of the model parameters (that aren’t done by learning algorithm). So the gradient-based RMSProp in Figs 2.c and 2.d isn’t involved in any kind of parameter manipulations. Moreover, the methods are compared for the same model parameters.
>
> To avoid confusion, we deleted the sentence: „… the same model parameters are used before weight updates in both methods”. Thanks for pointing that out!
>
> The adaptive learning rate for both methods (the first iteration of our algorithm and the rest of it) is kept the same. As Algorithm 2 says: $\theta \gets \theta + |\theta'-\theta|\cdot sgn(\theta.\mathrm{averagedGrad})$.
>
> Re. RD ISSUE 2: You also asked about the independence of the previous parameter updates. We don’t assume full independence, but we would have rather weaker assumptions: 1) every time, the direction of the current update of each parameter (+ or – only per parameter) is computed with the assumption to minimize the loss of a current batch and without the assumption to minimize the loss for any other batches, and 2) during computing RD the absolute value of the change of each parameter is equal for both methods (only signs may differ) (However we don’t write this assumptions in the paper). This assumption is satisfied for the vanilla RMSProp. Our paper for the reference: „$\mathcal{RD}$ would not be as useful when using momentum because the metric compares the aggregated loss of a single batch per parameter update, whereas momentum contributes to a decrease in loss over many batches per a single parameter update.” Please note, that the independence on the previous updates (that you wrote about) is impossible in any **practical** experiments, since the parameter values, (for which we compute RD), always depend on the previous updates (if no previous updates are, then we have only one random parameter values. We can have many random parameter values to compute RD, but this isn’t practical conditions).
>
> Maybe RD=10.41±1.94 looked strange, but it is so high because the gradient updates are done for the same learning rate (LR) as our algorithm, so the LR is too large for the gradient. It is pointed out in the Conclusions: „Moreover, it is crucial to note that the mentioned gain occurs at learning rates that are three times higher than the optimal rates for gradient-based training.”
>
> Still, we will update our RD paragraph to highlight its limitations, because, for example, a method that finds local minimums very efficiently may have high RD, but low accuracy and loss. That’s why accuracy and loss are conclusive.

---

### Official Review · Reviewer_t4AB · 2024-11-03

**Soundness:** 2
**Presentation:** 2
**Contribution:** 1
**Rating:** 3
**Confidence:** 3

**Summary:**

This paper presents a new workflow for training neural networks by modifying both the gradient calculation and the update recursions of standard algorithms. The authors replace the current gradient with an averaged gradient along the potential update direction, which is predefined by the exact gradient. The paper also introduces a cost-effective propagation method for calculating the average gradient. Small-scale experiments are conducted to support the proposed algorithm.

**Strengths:**

The paper offers a valuable perspective on leveraging explainable AI techniques, specifically, *Integrated Gradient* [1], which the authors refer to as the average gradient, to support neural network training.

>[1] Sundararajan, M., Taly, A., & Yan, Q. (2017, July). Axiomatic attribution for deep networks. In International Conference on Machine Learning (pp. 3319-3328). PMLR.

**Weaknesses:**

The main weakness of the paper is that computations based on average/integrated gradients are typically costly, as mentioned by the authors, but the effectiveness of the proposed average gradient approximation for acceleration remains uncertain. The average gradient propagation algorithm in Eq. (1) relies entirely on the rough approximation in Eq. (17). However, it is highly dependent on the properties of the decomposed integrand, and as the neural network structure becomes more complex, this assumption becomes increasingly unreasonable. Although the authors present Fig. 5 to illustrate their motivation, it is still not difficult to find counterexamples that challenge such an approximation.

The experiments are generally weak and not sufficiently convincing, involving only toy datasets such as MNIST/Fashion-MNIST and shallow networks. Given that the algorithm uses approximations during training, it is essential to carefully evaluate its effectiveness using more representative datasets and complex model architectures. Additionally, there may be issues with the experimental setup that have led to the claimed conclusions. Please refer to the questions section for further details.

The presentation of the algorithm in this paper is somewhat unconventional, and the connections between multiple algorithmic blocks are unclear. It would be helpful if the authors could present these blocks in a more concise and mathematical format, clearly specifying their outputs.

**Questions:**

- Why are there two rows in Tab. 3 with guessed learning rates, while a hyperparameter search was conducted for other identical models or algorithms? Moreover, the range of the search differs between the proposed algorithm and the baseline. Hyperparameter selection could be conducted more comprehensively, as multiple locally optimal learning rates are observed in Tab. 3.

- The proposed algorithm shows similar performance to the standard algorithm on Model A, but on Model B, the authors claim it is three times faster. Could this be because the chosen learning rate is precisely three times that of the baseline?

- The authors mentioned the improvement in computation time achieved by this method. It would be more informative if results were provided showing the relationship between the algorithm's wall-clock time cost and its accuracy.

---

> ### Author Response · Authors · 2024-11-17
> **Answers to the questions**
>
> 1: Firstly we conducted a hyperparameter search on Fashion MNIST. We got the optimal learning rate for the Average Gradient 3 times higher, so we made it 3 times higher on MNIST also as it resulted in very good results. We made more effort to optimize LR for vanilla RMSprop to give it an advantage. In general, for the vanilla RMSprop it was only a part of the hyperparameter search that is presented in the table, and **you are right, it is unclear**. The initial learning rate search was up to 0.0007 for vanilla RMSprop on MNIST and Fashion MNIST and the results for high learning rates were worse. Moreover, **during the review period we conducted much longer hyperparameter search (10 runs for every learning rate {$1.5e-4,2e-4,\ldots,5.5e-4$} on MNIST and {$2e-4,2.5e-4,\ldots,6e-4$} on Fashion MNIST), where the optimal hyperparameters are very similar and give nearly identical results: Model B on MNIST: $lr=3.5e-4\pm1.5e-4$ with min. loss over training $0.0856\pm 0.0139$ and Model B on Fashion MNIST: $lr=4e-4\pm1.5e-4$ with min. loss over training $0.318\pm0.016$**. These numbers are **nearly identical** to the minimal training losses observed in our experiments for the vanilla RMSProp. To ensure that there is no (unexpected) local minimum in the learning rate search space, we ran the ‘sanity check’ on high learning rates which proves that our method is \textit{much better**: on Fashion MNIST using LR=0.0009 (used for our algorithm), the vanilla RMSProp (**500** epochs) min. loss$=0.641\pm 0.168$ (10 runs), our 2 iter. (**300** epochs) min. loss$=0.257\pm 0.014$, our 5 iter. (**300** epochs) min. loss$=0.254\pm 0.017$. On MNIST using LR=0.0015 (2x higher than we finally used for our algorithm, but initially thought we would use the higher one), the vanilla RMSProp (**500** epochs) min. loss$=2.09\pm 0.05$ (10 runs), our 2 iter. (**300** epochs) min. loss$=0.397\pm 0.194$, our 5 iter. (**300** epochs) min. loss$=0.145\pm 0.036$. **We will add this information to the paper.** Moreover, as we mentioned in the paper, we made additional sanity check with trainings of our algorithm for the lower LRs that are optimal for the vanilla RMSProp, and our algorithm was already faster, but only some metrics were statistically significant (training counts were 2 and 3). Thank you for the **valuable insight**, we will make some modifications to make it clearer.
>
> 2: Due to the results given in the answer for your question no. 1, it is **certain** that the algorithm is **significantly** sample efficient.
>
> 3: Thanks for that insight. As we write in the paper, there are 2 possibilities for time comparison: a) plot according to our suboptimal implementation runtime, which would be deceiving, because our code can be optimized to reuse some of the backend variables more efficiently, some statistics don’t have to be computed (like loss change requires an additional inference per batch) and there are possible some low-level optimizations (C++ backend), b) we can assume how fast an optimal implementation can run, which is somewhere in between [2,2.333] times slower for 2 iterations than the vanilla RMSprop, but it might be awkward to assume a particular number, since it still would depend on some hardware details also to some extent. But we can do this, certainly such figure with proper comment would be very informative.

---

> ### Author Response · Authors · 2024-11-17
> **Answers to weaknesses and strengths**
>
> Answers to weaknesses (paragraphs are numbered):
> 1: **Certainly not. The relative time-per-epoch gain along sample efficiency is mentioned in both Results and Discussion section.** You have to **divide them** to get the relative difference of how long the loss is optimized to a certain level. Using 2 iterations and looking at median losses, our implementation is similar to the vanilla RMSprop to minimize loss to the certain level (3 times higher sample efficiency / 3 times slower per epoch), but **optimally it would be faster** (look at our answer to your third point in the questions).
>
> 1 continued: **Eq. 1 is proven in Appendix B**, and there aren’t assumed any „properties of the decomposed integrand” in the B3 part of the proof, which **proves approximation accuracy**. **The properties that you probably mean, that are assumed in B2, aren’t necessary in B3, and here is why:** B3 defines the accuracy of the approximation in Eq. 1, so \textit{the assumptions in B2 on the accuracy of the approximation aren’t needed at the point of B3, because B3 proves the accuracy**. That assumptions in B2 wouldn’t make any sense except to provide an intuition, because **they aren’t even precise**. You probably mean the text after „Rapid changes in the gradient (…)”. Moreover, you wrote about \textit{counterexamples**, but B3 proves that the **expected** accuracy of the approximation is better than for the gradient, so there may be specific counterexamples.
>
> 2: The same optimizers tend to perform well on small and big datasets. However, we will try to add one more deeper model and another benchmark before the deadline.
>
> 3: Thanks for that insight. We will do that soon. Indeed, it would be clearer if our subprocedure calls were outlined for example in bold. There will be many additional variables, like intermediate layer results or gradients, so we will think about the possibilities to keep it easily readable. For now, the pseudocode is roughly a simplified implementation in PyTorch, so it is designed for ease of implementation.
>
> Answers to strengths:
> We will add the citation. The average gradient is a very similar idea, but one of the differences is when the width of the range of integration approaches zero. Then the average gradient equals the gradient (so we still the have information about potential direction of loss minimization), so then our method turns into regular RMSprop for specific parameters during that particular updates. However, **we present a much, much faster algorithm (to obtain the integral) than [1] and prove the performance guarantees of its core equation, which are significant contributions**, and we argue that it is bigger contribution than [1].

---

### Official Review · Reviewer_xepv · 2024-11-04

**Soundness:** 2
**Presentation:** 1
**Contribution:** 2
**Rating:** 3
**Confidence:** 4

**Summary:**

This research paper introduces a novel algorithm for training deep neural networks with numerous nonlinearities. The proposed approach involves estimating the average gradient over the range of a potential parameter update, offering a method to enhance learning. In the implementation, the efficiently approximated average gradient is combined with RMSProp and evaluated against standard gradient-based techniques.

**Strengths:**

1. The underlying assumption of averaging is intriguing.
2. The observation in Figure 1 is noteworthy.

**Weaknesses:**

1. The paper is generally hard to follow, making it difficult to grasp the contribution and novelty of the work.

2. There is a significant amount of irrelevant information of the algorithm and writing typos. For example, several areas in the paper lack clarity:

   - In the abstract, there is *[...]* which is weird.
   - The start of the introduction is nearly a rephrasing of the abstract.
   - The introduction primarily repeats points from the general conclusion (e.g., RMSDrop performs well for deep models), diminishing the value of the information.
   - The introduction reads like a collection of related work, making it difficult to follow and affecting the logical flow.
   - Line $112$ contains a vague statement without added insight.
   - The figure provides limited information and is challenging to interpret.

3. There are three algorithms, all of which rely solely on descriptive text without any mathematical representation.

4. The algorithm is validated only on MNIST and Fashion MNIST data, which are relatively simple, toy examples for deep learning tasks.

**Questions:**

1. Could the algorithm section be condensed, as it's challenging to identify which algorithm is most significant?
2. How does the proposed algorithm relate to models with *many* nonlinear layers?
3. How does the algorithm perform on larger datasets, such as ImageNet?

---

> ### Author Response · Authors · 2024-11-17
>
> Answers to the questions:
> 1: We think that without any of the pseudocodes the method wouldn’t be easily reproducible without the code and unambiguous. However, all of the important knowledge is outside. We will try to make them clearer, however right now they refer to the flow that is characteristic to ML libraries like PyTorch, so at least a part of the readers would appreciate that. Especially those who will reimplement the algorithm. But we think about some changes to them.
>
> 2: As the Algorithm 3 and Equation 6 state, the backpropagation of the average gradient differs from the gradient backpropagation (base method) only by propagation through nonlinear activations (the implementation can be modified for all nonlinear functions, but in practical machine learning, at least nearly always, the only nonlinear operators are activations). Without nonlinear activations, our algorithm behaves exactly as the vanilla RMSProp. Therefore, the algorithm is for models with many nonlinear activations. We refer to Appendix E for further reading, it is about that.
>
> 3: We didn’t run on ImageNet for the following reason: The same top gradient-descent optimizers tend to perform well on small datasets, like MNIST, and the same for on larger datasets like ImageNet. Therefore, we would rather try other models in the first place, and by this occasion, they can be trained on variety of domains. So, to answer your question, we would expect the performance to be significantly more dependent on the model than on the dataset.  We hope **to train and include one deep model and dataset to the paper before the deadline**.
>
> Don’t hesitate to ask further questions! We are glad to answer.
>
> Answers to weaknesses:
> 1: Thanks for that note, we will clarify the contribution and novelty in the introduction. However, our contribution is already mentioned not only in the introduction but also in the abstract and conclusions: **the whole proof is ours, Eq. 1 and other equations are our contributions, whole approach of efficiently approximating the average gradient is also ours.** In short, this is extremely efficient algorithm version of [1] with the proof of performance guarantee of its core equation (eq 1).
>
> [1] Sundararajan, M., Taly, A., & Yan, Q. (2017, July). Axiomatic attribution for deep networks. In
> International Conference on Machine Learning (pp. 3319-3328). PMLR.
>
> 2: There are no writing typos. The ‘[…]’ is **due to the anonymization of the ICLR submissions.** Github links deanonymize.
>
> 2b: We will change the introduction, thanks for that note.
>
> 2c: The same as 2b.
>
> 2d: **No, only half of the second page is about related work.**
>
> 2e: **No.** Line 112: „All of the best and most popular optimizers for training large neural networks rely on the gradient.” Afterward, we write about the weakness of the gradient, that we solve. So no, it isn’t **„without added insight”.**
>
> 2f: The figure **do** provide a very important information. You can see in the figure, that from the average gradient, which is the direction coefficient of the presented linear functions, you can trivially retrieve the information about a change in the loss (100% accurately), for example using basic geometry. Moreover, it can be seen, that this doesn’t hold for the gradient. The information about the accurate change in the loss is very useful, as we explain in the paper.
>
> 3: **No**, the mathematical representation is fully presented as equations no. 1-6. The pseudocode refers to them in particular lines of code. The pseudocodes are mainly for the purpose of easier reimplementation, since they match a typical pytorch flow, and they are for optional clarifications of the details of the procedure. If we switch them to equations, we will have to repeat the equations no. 1-6. However, we aim to make the pseudocode flow more clear, as well as think about some other changes to them.
>
> 4: The answer is the same as for your question no. 3 (not weakness no. 3). Thanks to using those benchmarks, we could conduct more comprehensive hyperparameter search and achieve higher statistical significance in the results.

---

### Official Review · Reviewer_YTnu · 2024-11-04

**Soundness:** 2
**Presentation:** 2
**Contribution:** 1
**Rating:** 3
**Confidence:** 3

**Summary:**

The paper presents an algorithm using an average gradient to improve deep neural network training, paired with RMSProp. It claims better generalization, sample efficiency, and robustness across learning rates, with notable gains in deep models but limited improvements in shallow ones.

**Strengths:**

The motivation of the paper is clearly articulated, and based on the example in Figure 1, I believe that considering the use of average gradients is indeed a worthwhile alternative to traditional gradients. This paper makes an interesting attempt to explore the integration of concepts from another field (see Weaknesses part) into training.

**Weaknesses:**

1. Lack of contribution. The core idea of using average gradients is not new, as it closely aligns with the concept of "integrated gradient" in the literature. It is a direct shift of the integrated gradient [1] in the field of explainable AI.

> [1] Sundararajan, M., Taly, A., & Yan, Q. (2017, July). Axiomatic attribution for deep networks. In ICML.

2. The first main weakness of the paper is that computations based on average gradients are typically costly as mentioned by authors. If all gradients are replaced with average gradients, this would require calculating an integral for each neural network parameter in addition to its gradient at every iteration, resulting in an intractable complexity in practice. The authors also seem to have not reported the computation times in their experiments.

3. The theoretical foundation of the proposed average gradient approximation appears to be incorrect. The method proposed in this paper is based on the chain rule in Eq. (1). However, this chain rule for average gradients is incorrect, and it is easy to construct counterexamples. Looking at the proof, Eq. (17) is obviously incorrect; the integral of a product of functions does not equal the product of their integrals. Therefore, the authors must either provide a characterization of the class of integrand functions that satisfy Eq. (1) or measure the error of the chain rule approximation and demonstrate that its impact is negligible. Otherwise, despite the authors presenting some special cases, the theoretical foundation of this paper cannot be justified.

4. Experiments only contain toy examples. Both theoretically and empirically, there is a concern that the proposed method may not scale effectively to moderately larger problems. Furthermore, there is a notable absence of comparisons with established benchmarks.

**Questions:**

See Weaknesses.

---

> ### Author Response · Authors · 2024-11-17
>
> As you requested, here are answers to the weaknesses:
>
> 1: [1] presents a very **computationally ineffective algorithm** that relies on summation of gradients across many points of a predefined range (Riemann Sum). It is ineffective, because: a) it requires $n$ forward and backward propagations in $n$ points for the Riemann Sum of $n$ terms, therefore it **scales poorly, as the Riemann Sum algorithm does**. Our algorithm obtains very good convergence after 2 forward and backward propagations, which is **significantly better**. b) [1] algorithm does not change the range of integration during getting more and more precise approximation of integrated gradient. Changing the range of integration towards the direction pointed by already approximated average gradient is a feature of our method and enables us to iteratively find the approximated optimal update, where the approximation **is proved in appendix B3 to be more accurate than for the gradient**.
>
> In [1] the range of integration includes black image or zero vector, so it is a different approach. However, due to **our algorithm to approximate the integral is much much faster**, it can be plugged into their algorithm to speed it up. The next improvement is that for update magnitude approaching to zero the average gradient equals the gradient (so we still the have information about potential direction of loss minimization), so then our method turns into regular RMSprop for specific parameters during that particular updates.
>
> **Thanks for your insight**, we will clarify our contributions in the paper.
>
> 2: **Everything is completely opposite to what you wrote. Even in the abstract: „in  the
>  case of optimal implementation, learning would require less computation time than
>  the gradient-based RMSProp.”, while our SUBOPTIMAL implementation is approximately equal to RMSProp in that respect. Those are our results. Our algorithm is specifically designed to avoid the things that you wrote. The average gradients are calculated according to Eq. 1 which is VERY fast and we PROVED its properties in the appendix B.** **The comparison of times per epoch is repeated in the last paragraph of the results and in conclusions!**
>
> 3: Appendix B3 proves that the **expected** accuracy of approximation in Eq. 1 is better than for the gradient, so there may be specific counterexamples! But the proof is for a general case which is defined and indeed is general.
>
> **WE DIDN’T WRITE THAT: ‘the integral of a product of functions does equal the product of their integrals’. WE USED APPROXIMATION SIGN ‘≈’ and WE PROVED THE ACCURACY OF APPROXIMATION.** (Appendix B3). Please ask questions instead of statements if something needs clarification.
>
> In the Eq. 17, which you mention, we wrote the approximation sign (‘≈’), **not equality that you wrote**, and at that point of the proof we don’t specify the accuracy of the approximation. The accuracy is proven later in appendix B3.
>
> 4: We will try to add one more practical very deep model on completely different benchmark. Hopefully, we will do this before the deadline. We have also to add, that the same top gradient-descent optimizers tend to perform well on small datasets, like MNIST, and the same for on larger datasets. Therefore we would rather try other models in the first place, and by this occasion they can be trained on a variety of domains. As we wrote in Appendix E, the performance depends rather on a model – in general, the more nonlinear activations, the better.
>
> Please ask questions for further clarification.

---

> > ### Comment · Reviewer_YTnu · 2024-11-30
> > **Response to rebuttal**
> >
> > Thank you for your rebuttal. I have decided to maintain my assessment.

---

### Author Response · Authors · 2024-11-17
**Why do we think that the community is interested in our work?**

We would argue that the community is interested in the fact, that **a modification of gradient on nonlinear activations of very deep models may increase sample efficiency multiple times**, which is the direct conclusion of our experiments.

We think that the community could be interested in the results on model B because it shows, that not all quite simple models with common activation functions (that are the most similar to biological activations working in ‘binary fashion’) can be trained with vanilla gradient, which is proven in our experiments to be far from the optimal algorithm to minimize the loss. Human brain has many thousands of neural layers, while the RMSprop algorithm fails to efficiently solve 30 layers of dense! This indicates, that the biological brain works in a distinct manner than pure gradient descent. We **prove and implement** a possible solution, that works for sure in some cases at least. Importantly, the empirically proven huge inoptimality of the vanilla gradient means, that historical ML-architecture searches were biased towards models with good behavior when trained with inoptimal training algorithm. Example: a lot of very deep models are trained much more efficiently and achieve better scores with residual connections, while if gradient descent was optimal to minimize the loss, many of them could **be trained** to behave in such manner, that the intermediate-layer data is passed through further layers, without skip connections. Therefore, the residual connections make huge difference to the vanilla gradient. Therefore historical ML-architecture searches based on gradient optimizers were biased towards residual connections and shallow models. Our experiment shows that, at least in some cases, the inoptimality of learning can be solved efficiently.

It is also worth mentioning that if the sample efficiency according to median training loss increased three times for Model B, and even if the gain was 50 times smaller for some different very deep models (which is unlikely to happen for the same type of layers and activations), it would be still **significant achievement** for our algorithm. Also similar optimizers work well across domains, so a domain switch isn’t expected to bring significant degradation of the results. However, \text{we will try to finish hyperparameter search and tests on one more benchmark and a more practical model architecture until the deadline.**

Moreover, we present a much, much faster algorithm than [1] and prove the performance guarantees of its core equation (Eq. 1), therefore our contribution is much bigger than [1].

[1] Sundararajan, M., Taly, A., & Yan, Q. (2017, July). Axiomatic attribution for deep networks. In International Conference on Machine Learning (pp. 3319-3328). PMLR.

**We also plan to add one more model and dataset, hopefully before the deadline.**

---

### Author Response · Authors · 2024-11-20
**Recent Changes**

- More results added to the hyperparameter-search table.
- Fig. 1 modified for clarity - plot and caption modifications (the marks are clearly visible, each subplot contains a short description which makes it easier to understand, local minimums are marked on the legends, now it is clear that both of the subplots should be interpreted separately and we changed ‘After Parameter Update’ on the legend to ‘After Parameter Update Suggested by the Gradient’).
- Added four more literature references and two sentences about the literature.
- One very important future work research direction added.
- Our functions in the pseudocodes are marked bold for the clarity dependencies in the pseudocodes
- Deleted one sentence in the RD paragraph.

---

### Author Response · Authors · 2024-11-28
**Important Changes**

More results are added, new IMDB trainings demonstrate excellent performance across various domains and one more deep model:
- We **added results of a very deep convolutional neural network with Tanh activations on the IMDB dataset**, to verify the performance on NLP and a model primarily based on convolutional layers (Appendix H). **Sample efficiency is approximately 155% of that achieved by gradient-based RMSProp, using only 2 iterations of our method.**
- **Included experiments with Model B using alternative weight initialization** to verify robustness across different weight initialization distributions (Appendix I). **Our algorithm is more than 2.5 times faster at minimizing mean loss in these experiments.**

All other issues have been addressed (some ideas were not incorporated into our paper, but we commented on them clearly):
- Completely revised the first subsection of the introduction, removed duplication of information.
- Added a sentence in the caption of Fig. 1 to clarify the relation of the illustrations to the n-dimensional case.
- Included a short note about the purpose of the RD metric in the paragraph discussing RD in the Methods section.
- Emphasized that the assumptions in the paragraph before Eq. 18 are included solely to build intuition about the approximation in Equation 18. These assumptions are not required in the proof, as the proof of the precision of approximation in Eq. 1 is separate.
- Added the definition of 'optimal implementation' at its first occurrence in the main body of the paper.
- Added a reference to the proof in the Appendix B near Equation 1.
- Mentioning in the conclusions that the RD metric values validate the proof in Appendix B, as both focus on the efficiency of batch-loss minimization.
- Using '≅' instead of '≈' for approximate equality in most places (**no change in the meaning** of the equations).
- Moved nearly all future work directions to the Appendix J.
- Made a few tweaks to the discussion and conclusions.

We would like to thank the reviewers for the valuable comments, which helped us refine the paper!

---

### Author Response · Authors · 2024-11-28
**Why We Did Not Test SOTA Models and Benchmarks Bigger Than 60K Examples**

At the moment, the only argument against our paper that needs clarification is the issue of testing our algorithm. We added a lot of additional results yesterday, and we argue that they are sufficient to prove our point, along with our proof in Appendix B. We did not evaluate our algorithm on state-of-the-art models using large datasets for the following **important** reasons:
- SOTA models are designed to work well with gradient descent because they are intended to be trained with gradient descent. For example, gradient descent does not perform well on very deep architectures without skip connections [1]. Therefore, SOTA models utilize many residual connections, causing such models to rely on simple features [2], mimicking an ensemble of shallower models [2]. We primarily want to test our algorithm in scenarios where the learning task cannot be **simplified** to the equivalent of training shallow models, which is easy.
- The same training algorithms tend to work well for both small and large datasets and models, such as RMSProp and Adam.
- Our Python implementation is limited to sequential models without dropout or skip connections because we had to implement a computation dependency graph for the average gradient from scratch in Python. The programming work was very challenging and time-consuming because we wanted to ensure compatibility with PyTorch, which has counters for gradient-tensor modifications and halts execution if it detects mismatched counters (with no way to turn this off). So the programming work was akin to hacking to bypass the 'safety' mechanisms of the PyTorch library (there was no simpler solution). To test our algorithms on SOTA models, we would have to code a significant portion of PyTorch or TensorFlow from scratch, which would be extremely time-consuming.
- A large number of training runs (i.e., **10**) would be needed to detect a sample efficiency gain of around, for example, 15% with high significance. In the case of backpropagation of the average gradient through nonlinear activations on large feature maps, our implementation with two iterations could be up to **4** times slower (this is not the case with Model B). Therefore, the total computational effort required for such an experiment would be up to **20×4=80** times higher than in some SOTA papers. (The numbers provided are just to indicate the order of magnitude.)

All of the above arguments collectively justify our decision to test the models we selected. We contend that our arguments are sufficient.

Importantly, our algorithm **solves** the **shattered gradients problem** as described in [1], which is a significant contribution. Computing the accurate influence on loss for each model parameter reduces the white gradient noise described in [1]. (We proved that we compute the accurate influence on loss in Appendix B and demonstrated it in our experiments.)

[1] Balduzzi, David, et al. "The shattered gradients problem: If resnets are the answer, then what is the question?." International conference on machine learning. PMLR, 2017.
[2] Veit, Andreas, Michael J. Wilber, and Serge Belongie. "Residual networks behave like ensembles of relatively shallow networks." Advances in neural information processing systems 29 (2016).

---

### Meta-Review · Area_Chair_N2CC · 2024-12-16

**Metareview:**

The paper presents an algorithm for training deep neural networks using the “average gradient" rather than the standard gradient. The average gradient is an average of the parameter gradients over a potential weight update — which the authors claim should perform better.

The reviewers found the ideas interesting and the presentation clear. They also found the work to have limited empirical validation, concerns about the cost and scalability of the method, and overall had issues with the paper’s presentation.

I agree with the reviewers that the scope of experiments is lacking by today’s standards. I do not think even imagenet is necessary but in my experience I’ve found many things that work well on mnist/fashion-mnist can fail at even small image datasets like cifar10. As well, the method presented requires more compute than standard SGD. The authors claim that their method reaches the same loss of SGD in fewer steps but there are other ways to trade off gradient steps for model compute such as (for example) increasing model parameters. I would want a more detailed analysis on how effectively the method improves compute efficiency.

Overall I believe the idea is interesting but I would want a more detailed empirical study to recommend acceptance.

**Additional Comments On Reviewer Discussion:**

Initially the reviewers had negative sentiment about the work especially around the paper's clarify and experiments. The authors provided some new experiments to respond to reviewers which reviewers found unconvincing. The reviewers generally felt that the authors' tone was combative and this stalled discussions. The authors did not sufficiently address the reviewer's main concerns regarding the toy-ish experiments and computational burdens of the method and chose to provide explanations for the lack of these experiments (which the reviewers did not agree with).

---

### Decision · Program_Chairs · 2025-01-22

Reject